# The WT1-like transcription factor *Klumpfuss* maintains lineage commitment of enterocyte progenitors in the *Drosophila* intestine

Jerome Korzelius[1,4], Sina Azami [1,4], Tal Ronnen-Oron[2], Philipp Koch [1], Maik Baldauf[1], Elke Meier[1], Imilce A. Rodriguez-Fernandez[3], Marco Groth [1], Pedro Sousa-Victor[2] & Heinrich Jasper[1,2,3]

In adult epithelial stem cell lineages, the precise differentiation of daughter cells is critical to maintain tissue homeostasis. Notch signaling controls the choice between absorptive and entero-endocrine cell differentiation in both the mammalian small intestine and the *Drosophila* midgut, yet how Notch promotes lineage restriction remains unclear. Here, we describe a role for the transcription factor Klumpfuss (Klu) in restricting the fate of enteroblasts (EBs) in the *Drosophila* intestine. Klu is induced in Notch-positive EBs and its activity restricts cell fate towards the enterocyte (EC) lineage. Transcriptomics and DamID profiling show that Klu suppresses enteroendocrine (EE) fate by repressing the action of the proneural gene Scute, which is essential for EE differentiation. Loss of Klu results in differentiation of EBs into EE cells. Our findings provide mechanistic insight into how lineage commitment in progenitor cell differentiation can be ensured downstream of initial specification cues.

[1] Leibniz Institute on Aging–Fritz Lipmann Institute (FLI), Jena, Germany. [2] Buck Institute for Research on Aging, 8001 Redwood Boulevard, Novato, CA 94945-1400, USA. [3] Immunology Discovery, Genentech, Inc., 1 DNA Way, South San Francisco, CA 94080, USA. [4] Present address: Max-Planck-Institute for Biology of Aging, Cologne, Germany. Correspondence and requests for materials should be addressed to J.K. (email: jkorzelius@age.mpg.de) or to H.J. (email: jasper.heinrich@gene.com)

In many tissues, somatic stem cells respond to tissue injury by increasing their proliferative potential and producing new differentiating cell progeny. To maintain homeostasis during such periods of regeneration, cell specification and differentiation need to be precisely coordinated within a dynamic environment. Studies in the mammalian intestine have demonstrated a surprising plasticity in such specification events, showing that even differentiated cells can revert into a stem cell state during times in which tissue homeostasis is perturbed[1,2]. These findings highlight the critical role of gene regulatory networks in establishing and maintaining differentiated and committed cell states in homeostatic conditions.

The *Drosophila* midgut is an excellent model to study lineage differentiation of adult stem cells both in homeostasis as well as during regeneration and aging. The *Drosophila* midgut is maintained by intestinal stem cells (ISCs), which can generate differentiated enterocytes (EC) or enteroendocrine (EE) cells[3,4]. Upon injury or infection, ISC proliferation is dramatically increased in response to mitogenic signals from damaged enterocytes[5–7]. Misregulation of cell specification and differentiation in this lineage can lead to substantial dysfunction, as evidenced in aging intestines, where disruption of normal Notch signaling due to elevated Jun-N-terminal Kinase (JNK) signaling leads to an accumulation of mis-differentiated cells that contribute to epithelial dysplasia and barrier dysfunction[8,9].

Notch signaling plays a central role in both ISC proliferation and lineage differentiation. ISCs produce the Notch-ligand Delta and activate Notch in the enteroblast (EB) daughter cell. This Notch-positive EB is the precursor of mature enterocytes (ECs). Levels of Delta vary markedly between ISCs in the homeostatic intestine. These differences have been proposed to underlie the decision between EC and EE differentiation in the ISC lineage:[10] high Dl-N signaling activity between the stem cell and its daughter is associated with EC differentiation, while lower Dl-N signaling activity between the ISC and its daughter promotes EE differentiation[10,11]. Loss of Notch in ISC lineages leads to the formation of tumors that consist of highly Delta-expressing ISCs and of Prospero (Pros)-expressing EEs[10,12,13]. These tumors are likely a consequence of impaired EB differentiation, resulting in an increased frequency of symmetric divisions, as well as excess EE differentiation, suggesting that EE differentiation is the default state when Notch signaling activity is absent or reduced.

Interestingly, recent work has shown that lineage specification in ISC daughter cells is likely more complex than previously thought. It has been shown that ISCs exist that express the EE marker Prospero and generate daughter cells that differentiate into EEs[14,15]. A transient specification step has been identified in EE differentiation, in which cells transiently express Scute, a transcription factor that negatively regulates Notch-responsive genes such as Enhancer of Split-m8 (*E(Spl)m8*), as well as its own expression[16]. Furthermore, EBs have been shown to remain in a transient state for a prolonged period of time before differentiating into an EC fate[17]. The exact cell state in which the decision between EE and EC fates is cemented, however, remains unclear.

Here we describe a role for the transcription factor Klumpfuss (Klu) in lineage commitment during EC differentiation in the adult fly intestine. Klu is related to the mammalian tumor-suppressor gene Wilms' Tumor 1 (WT1), and its overexpression in neuroblast stem cells leads to tumorous overgrowths in the brain of flies[18–20]. In the intestine, we find Klu to be expressed specifically in EBs. Loss of Klu leads to aberrant EE differentiation of EB cells, whereas ectopic activation of Klu results in a failure to differentiate. Transcriptomics and DNA-binding studies reveal that Klu controls EE differentiation by repressing genes involved in Notch signaling, as well as by indirectly controlling the levels of

the Achaete-Scute complex members *asense* and *scute*. Klu acts in a negative feedback loop by regulating its own expression and the expression of Notch target genes. We propose that Klu defines a transient state of EBs in which specification into ECs is cemented by precise regulation of Notch signaling: the expression of Klu locks in the EC fate in EBs by preventing ectopic proneural gene activation and thus ensuring lineage commitment into the EC fate.

## Results

**Klu is expressed in the enteroblast precursor cells**. We identified Klumpfuss (Klu) transcripts to be significantly downregulated upon loss of the stem and progenitor specific transcription factor Escargot (Esg) and to be enriched in transcriptomes of sorted Esg-positive (Esg+) cells[21,22]. To confirm *klu* expression in the *Drosophila* posterior midgut, we used a *klu-Gal4, UAS-GFP* reporter line that reflects Klu expression in wing and eye discs of wandering third instar larvae[23,24]. In the midgut, GFP expression was seen in the larger cells of the stem-progenitor nests (ISC+EB) and resembled EBs based on both nuclear and cellular size (Fig. 1a–c arrowheads). To confirm their identity, we combined the *klu-Gal4, UAS-GFP* line with the Notch activity reporter *Su(H)GBE-lacZ*, which is exclusively activated in EBs[10]. In addition, we used *Delta-lacZ (Dl-lacZ)* as a marker for ISCs. The expression of *klu-Gal4, UAS-GFP* overlapped almost exclusively with *Su(H)GBE-lacZ*. In contrast, *Dl-lacZ* staining was mostly found in small, diploid cells neighboring the GFP-positive cells (Fig. 1d–i, quantification in j, k). We confirmed the EB-specific expression of the enhancer-trap line by performing a knock-in replacement of the Klu Coding Sequence (CDS) with the Gal4 CDS (Supplementary Fig. 1, see Methods). To further confirm the expression of Klu in EBs, we used a FISH-probe for *klu* mRNA: this labeled *klu* mRNA in *Su(H)GBE-Gal4>UAS-GFP* marked EBs (Supplementary Fig. 1h, i, arrows).

Lineage-tracing experiments have previously shown that Notch-positive EB precursor cells exclusively give rise to enterocytes, whereas Delta-positive ISCs can give rise to clones with both ECs and EEs[14,15]. To trace the fate of Klu-expressing cells, we crossed the *klu-Gal4* enhancer-trap line to a Actin promoter-driven FlipOut (F/O) lineage-tracing cassette (*UAS-GFP, tub-Gal80ts; UAS-Flp, Act >STOP> Gal4*). As expected, *Dl-Gal4*-expressing ISCs gave rise to both ECs as well as EEs, marked by expression of the transcription factor Prospero (Pros) (Fig. 1l, m, arrows). In contrast, Notch-positive EBs (*Su(H)GBE-Gal4*) only gave rise to ECs, but not EEs (Fig. 1n, o, arrowheads). Similar to Notch-positive EBs, *klu-Gal4*-traced cells gave rise exclusively to ECs, but not EEs (Fig. 1p, q). We conclude that Klu is expressed in the EC-generating EBs in the *Drosophila* midgut.

**Klu loss of function leads to excess EE differentiation**. To determine the role of Klu in the specification and/or differentiation of cells in the ISC lineage, we first inhibited Klu function using the temperature-inducible TARGET-system to express RNAi constructs in specific lineages[25]. We used *esg-Gal4ts* to express *kluRNAi* in ISCs and EBs, and *Su(H)GBE-Gal4ts* to express *kluRNAi* in EBs only. In both conditions, knockdown of Klu increased EE numbers in the posterior midgut (Fig. 2a–d, quantification in Fig. 2i), suggesting that knockdown of Klu in EBs promoted the adoption of EE over EC fates in these cells. To confirm this, we used EB-specific FlipOut lineage tracing in combination with *kluRNAi* to trace the fate of *kluRNAi*-expressing EBs. We induced clones for 10 days at 29 °C, followed by a short 16-hour infection with the pathogen *Erwinia carotovora* (*Ecc15*) to induce gut turnover. Pros-positive EEs are seldom found in such EB-derived *Su(H)GBE-F/O* clones in control backgrounds, yet we found a significant increase of such

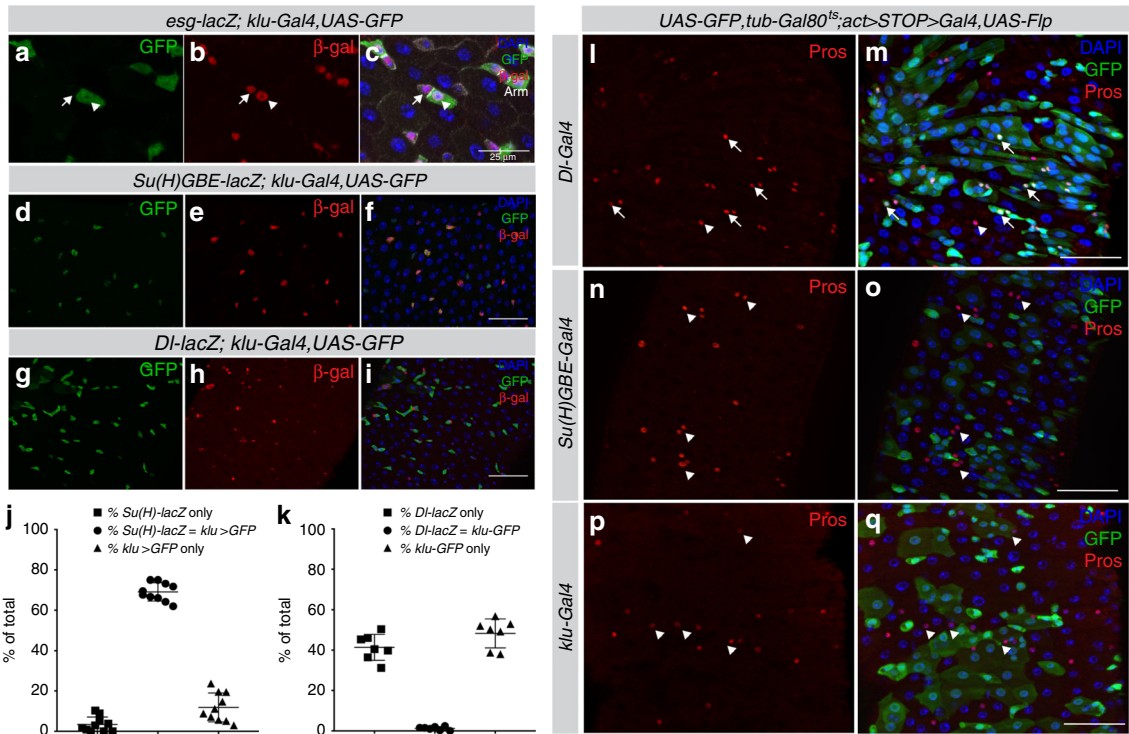

**Fig. 1** Klu is specifically expressed in enteroblast cells. **a–c** The *klu-Gal4, UAS-GFP* reporter line shows expression in the midgut epithelium. ISCs (arrows) and EBs (arrowheads) are visualized by *esg-lacZ* (beta-galactosidase, red). Cells are outlined with Armadillo/beta-catenin (Arm, grayscale). Representative area of posterior midgut is shown. *n* = 3 animals. **d–i** The *klu-Gal4, UAS-GFP* was combined with *Su(H)-GBE-lacZ* (enteroblast (EB) marker) or *Dl-lacZ* (intestinal stem cell (ISC) marker). Expression of Klu largely overlaps with the EB marker *Su(H)-GBE-lacZ* (**d–f**), and Klu-positive cells are found adjacent to the Delta-positive ISCs (**g–i**). **j** Quantification of EB-marker gene overlap of the genotypes displayed in **d–f**. *n* = 10 guts/animals (*Su(H)-GBE-lacZ*, *n* = 572 cells counted). **k** Quantification of ISC-marker gene overlap of the genotypes displayed in **g–i**. *n* = 7 guts/animals (*Dl-lacZ*, *n* = 1370 cells counted). **l–q** Lineage-tracing of cells in the intestine using different cell-specific drivers. EEs are marked by antibody staining for the transcription factor Prospero (Pros, red). Arrows indicate GFP-Pros double-positive EEs in the clonal area, whereas arrowheads indicate EEs outside the clonal area. **l, m** The *Dl-Gal4*-positive ISCs give rise to both differentiated cell types of the intestinal lineage (enterocytes (EC) and enteroendocrine (EE) cells). **n, o** *Su(H)-GBE*-positive EB cells exclusively give rise to ECs, but not to EEs. **p, q** Similar to *Su(H)-GBE*-positive EBs, *klu-Gal4*-positive cells give rise exclusively to ECs. Representative areas of posterior midgut are shown. *n* = 7 guts examined for **l, m**, *n* = 7 guts examined for **n, o** and *n* = 10 guts examined for **p, q**. Scale bar = 50 μm, except in **a–c**: scale bar is 25 μm

cells in clones expressing *klu^RNAi^* (Fig. 2e, f, quantification in Fig. 2j). To further confirm these results, we generated GFP-marked clones homozygous for a null allele of Klu, *klu^R51^* using the MARCM technique[24,26] and quantified EE numbers. Quantification showed that *klu^R51^* MARCM clones had more EE cells/clone (Fig. 2g, h, quantification in Fig. 2k). Interestingly, the GFP-negative tissue also contained more EEs in *klu^R51^* MARCM animals than in control animals (*FRT2A*, Fig. 2g, compare with Fig. 2h). This is likely due to the fact that in this genotype, the GFP-negative tissue is heterozygous for *klu^R51^*. Accordingly, MARCM RNAi (*FRT40A; klu^RNAi^*) clones (in which the surrounding tissue is wild type for Klu) had an increase in the number of EE cells/clone, but no difference in EE cells in the non-clonal surrounding tissue (Supplementary Fig. 2g, h, quantification in Supplementary Fig. 2i, j). These results strongly suggest that Klu acts cell-autonomously in preventing EE differentiation of EB.

Interestingly, EB-to-EC differentiation could still occur in *klu*-deficient lineages: *esg-F/O* clones expressing *klu^RNAi^* (*esg-F/O>klu^RNAi^*) still contained cells with large nuclear size and positive for the EC marker Pdm1 (refs. [21,27]) (Supplementary Fig. 2a–f). In summary, our results indicate that loss of *klu* alters the EE-to-EC ratio in ISC lineages, but does not fully impair EC differentiation.

**Ectopic Klu blocks proliferation and EB differentiation.** Based on these observations, we hypothesized that constitutive Klu

overexpression could reduce EE differentiation in the ISC lineage and might trigger ectopic differentiation of ISCs into ECs. To test this, we used the *esg-F/O* system to express full-length Klu in ISC-derived clones. Wild-type *esg-F/O* clones take up most of the posterior midgut 2 weeks after induction, containing a mixture of ECs and EEs (Fig. 3a). In contrast, clones expressing full-length Klu remained very small, containing only a few cells that did not exhibit any hallmarks of differentiation into either EEs or ECs (Fig. 3b). Klu is thought to act mainly as a repressor of transcription based on studies in other organs[18,23,28]. To ask whether this repressor function of Klu would elicit the phenotypes observed, we expressed the zinc-finger DNA-binding domain of Klu fused to either a VP16 activation domain (Klu-VP16) or fused to the repressor domain from Engrailed (Klu-ERD)[28]. Whereas clones grew normally and differentiation still occurred in clones expressing the activating Klu-VP16, clone size was smaller and differentiated cells were not observed in clones expressing the repressing Klu-ERD, confirming that transcriptional repression of genes regulated by Klu is sufficient to limit growth of ISC-derived clones (Fig. 3c, d, quantification in Fig. 3e). Similarly, *UAS-klu* expression in *esg-F/O* clones inhibited proliferation of ISCs (measured by quantifying mitotic figures in the gut) both in homeostatic and infected conditions (infection with *Ecc15*; Supplementary Fig. 3a). Restriction of Klu expression solely to ISC (using *esg^ts^* combined with *Su(H)-Gal80* (ref. [29]) showed that the repression of mitosis upon *Ecc15*

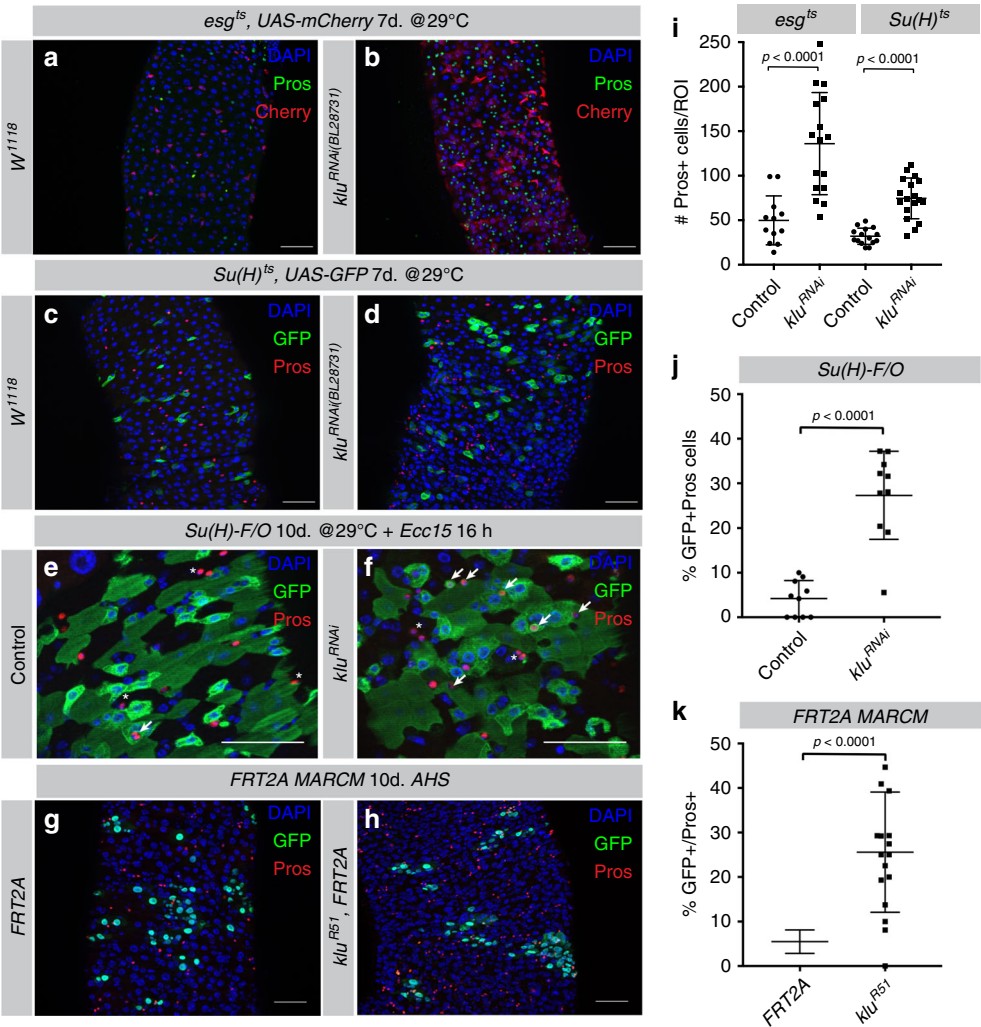

**Fig. 2** Loss of Klu leads to excess EE differentiation. **a–d** RNAi-mediated knockdown of Klu results in an excess of Pros-positive EE cells. Expression of $klu^{RNAi}$ using the ISC + EB driver $esg\text{-}Gal4^{ts}$ (Pros in green, compare **a** with **b**) or the EB-specific $Su(H)GBE\text{-}Gal4^{ts}$ driver (Pros in red, compare **c** with **d**). **e**, **f** $Su(H)\text{-}GBE$-driven FlipOut ($Su(H)\text{-}F/O$) clones expressing $klu^{RNAi}$ show an increased number of Pros-positive EE cells in the clonal area upon $Ecc15$ infection compared to controls (**e** compare with **f** quantification in **j**). **g**, **h** Clonal analysis of control $FRT2A$ (**g**) or $FRT2A$, $klu^{R51}$ (**h**) null mutant MARCM clones. Representative areas of posterior midguts are shown. **i** EE cell quantification of the posterior midgut for the genotypes in **a–d**. Number of midguts $n = 12$ (control $w^{1118}$) and $n = 16$ ($klu^{RNAi}$) for **a** and **b** and $n = 15$ (control $w^{1118}$) and $n = 18$ ($klu^{RNAi}$) in **c** and **d**. **j** Quantification of GFP-Pros double-positive cells/ROI in control and $klu^{RNAi}$-expressing $Su(H)\text{-}F/O$ clones in **e** and **f**. $n = 10$ for control and $n = 10$ for $klu^{RNAi}$ guts. **k** Quantification of the number of Pros-positive EEs/clone and the total number of Pros-positive EEs/ROI for the genotypes in **g** and **h**. $n = 15$ guts ($FRT2A$ control) and $n = 17$ guts ($klu^{R51}$). For quantifications in **i–k**: error bars represent mean ± SD. Significance was calculated using Student's $t$-test with Welch's correction. Scale bar = 50 μm

infection is mainly due to the ectopic expression of Klu in ISCs, although we do observe a small but significant decrease if we express Klu using the EB-driver $Su(H)^{ts}$ (Supplementary Fig. 3b). We also combined expression of Klu ($UAS\text{-}klu$) with expression of the oncogenic $Ras^{V12}$ variant ($UAS\text{-}Ras^{V12}$) in $esg\text{-}F/O$ clones. Whereas $esg\text{-}F/O>Ras^{V12}$ clones occupy the entire posterior midgut 2 days after induction and contribute to a rapid loss of viability of the animal, co-expression of $UAS\text{-}klu$ markedly reduced clonal size and rescued viability (Supplementary Fig. 3c–g). This is consistent with an anti-mitotic effect of ectopic Klu expression in ISCs.

To ask whether sustained expression of Klu in EBs would influence their differentiation, we performed lineage-tracing initiated from EBs. Indeed, continuously expressing Klu in EBs using $Su(H)\text{-}F/O >UAS\text{-}klu$ impaired the formation of differentiated Pdm1-positive enterocytes (Fig. 3f–i, compare with Fig. 3j–m, quantification in Fig. 3n). While ectopic expression in ISCs thus impairs proliferation, sustained expression of Klu in

EBs impairs EC differentiation. These results support a model in which Klu acts in early EBs to restrict EE differentiation, but it needs to be suppressed to allow EC differentiation.

To further characterize the gain-of-function phenotype, we combined $UAS\text{-}klu$ with the ISC-marker $Dl\text{-}lacZ$ and the EB-marker $Su(H)GBE\text{-}lacZ$. Interestingly, $esg\text{-}F/O$ clones expressing $UAS\text{-}klu$ did not stain positive for either $Dl\text{-}lacZ$ (Fig. 4a–d) or $Su(H)GBE\text{-}lacZ$ (Fig. 4e–h), suggesting that ectopic Klu expression in ISCs interferes with normal Dl-Notch signaling in ISC-EB pairs. To investigate this interaction between Notch signaling and Klu activity further, we performed epistasis experiments: Klu overexpression prevented the formation of large tumors in Notch loss of function $esg\text{-}F/O$ clones (Supplementary Fig. 4a–l) and $UAS\text{-}klu$ can repress the excess mitosis seen in the $esg^{ts}>N^{RNAi}$ genotype (Fig. 4i), consistent with the inhibition of ISC proliferation upon Klu expression.

To test whether Notch is required for Klu expression in EBs, we performed qRT-PCR for $klu$ on progenitor cells expressing

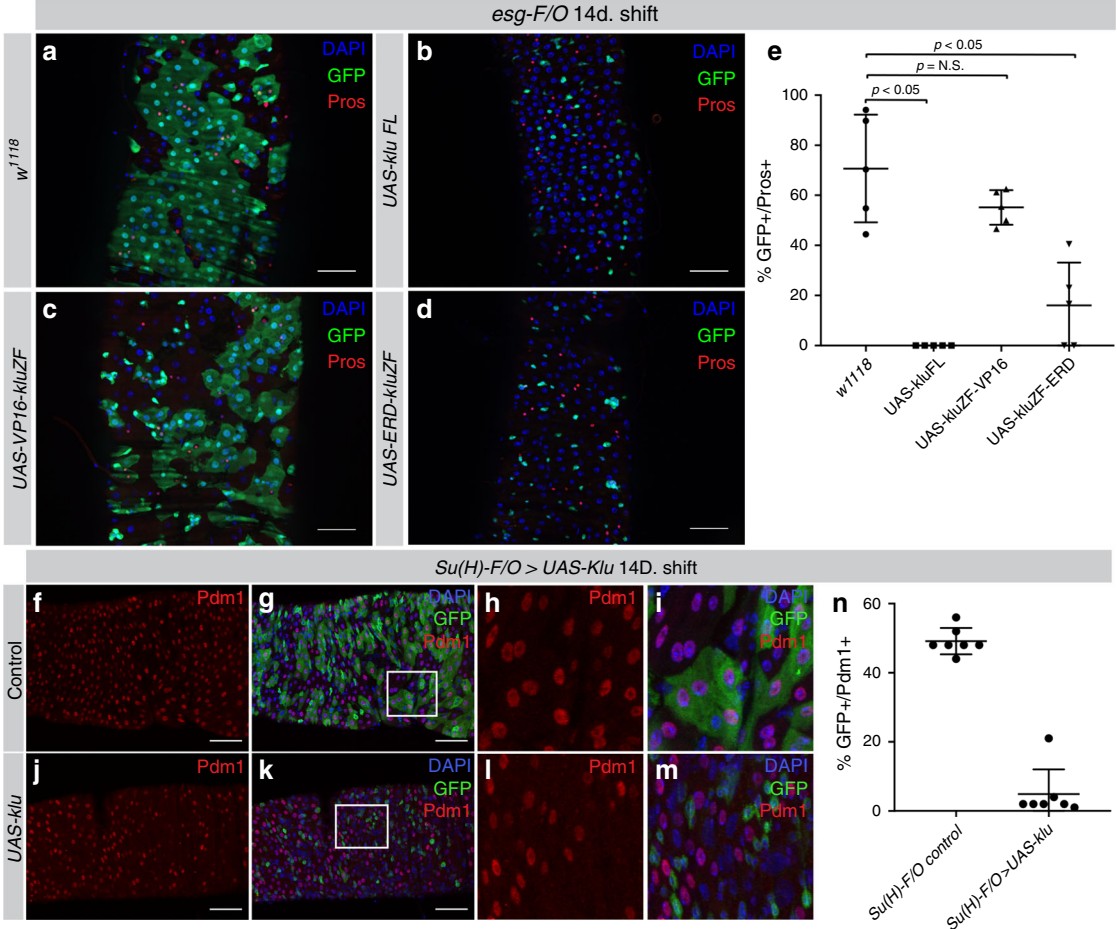

**Fig. 3** Klu overactivation results in a loss of Delta-Notch signaling and ISC differentiation. **a–d** Clonal expression of different Klu isoforms using the *esg-Gal4*-driven FlipOut (*esg-F/O*) system to generate ISC clones. **a** Control *esg-FO* clones grow to occupy most of the posterior midgut 2 weeks after clonal induction. **b–d** Clones expressing either full-length Klu (*UAS-kluFL*, **b**) or the Klu zinc-finger DNA-binding domain fused to the Engrailed Repressor Domain (*UAS-ERD-kluZF*, **d**) resulted in a block of differentiation. This was not observed when expressing the Klu zinc-finger DNA-binding domain fused to the VP16 transcriptional activator domain (*UAS-VP16-kluZF*, **c**). Representative areas of posterior midguts are shown. **e** Quantification of genotypes in **a–d**. $n = 5$ midguts for each genotype. **f–i** *Su(H)-F/O* control clones contain GFP-Pdm1 double-positive cells, representative of EB > EC differentiation (**f**, **g** closeup in **h**, **i**). **j–m** *Su(H)-F/O>UAS-klu* clones contained much less GFP-Pdm1 double-positive cells, indicative of impaired EB > EC differentiation upon Klu expression. **n** Quantification of the percentage of GFP-Pdm1 double-positive cells in images of posterior midguts from control (**f–i**) and *UAS-klu Su(H)-F/O* (**j–m**) clones. $n = 7$ midguts for each genotype. For quantifications in **e** and **n**: Error bars represent mean ± SD. Significance was calculated using Student's *t*-test with Welch's correction. Scale bar = 50 μm

$N^{RNAi}$ for 1 week ($esg^{ts} > N^{RNAi}$). Consistent with the formation of Pros$^+$ cell tumors, loss of Notch leads to a 5.5-fold upregulation of *pros* mRNA in Esg$^+$ cells. However, *klu* expression is almost absent from $N^{RNAi}$ cells (Fig. 4j), strongly suggesting that Klu expression depends on Notch activity.

Ectopic activation of Notch in stem-progenitor cells using the Intracellular domain of Notch ($esg^{ts} > UAS-N^{ICD}$) results in a loss of the stem-progenitor compartment due to premature differentiation into EC cells[10]. $UAS-N^{ICD}$ expression resulted in *klu* mRNA expression in large Esg$^+$ cells that seem to be differentiating into ECs based on their nuclear size (Fig. 4m, n, compare with Fig. 4k, l), suggesting that Notch activation is sufficient to induce Klu expression. However, combining $UAS-N^{ICD}$ with $klu^{RNAi}$ did not alter the premature differentiation phenotype of $UAS-N^{ICD}$ (Supplementary Fig. 4m–t).

Since Notch activation is thus sufficient to induce differentiation of Esg$^+$ progenitors into ECs even in the absence of Klu, we conclude that induction of Klu by Notch in EBs is important to prevent specification of EBs into EE progenitors, but is not essential for other steps in EC differentiation.

Altogether, our results indicate that the Notch-mediated induction of Klu in EBs is required to restrict lineage commitment of EBs to the EC fate. Reciprocally, ectopic Klu expression interferes with normal Delta-Notch signaling between ISC and EB and inhibits proliferation. We propose that ISC-derived EB daughter cells that express Klu enter a cell cycle arrested, undifferentiated state, and that Klu needs to be downregulated for EC differentiation to proceed. To test this hypothesis, and to understand how Klu expression controls the EB state, we decided to explore the transcriptional program downstream of Klu.

**RNA-Seq supports role of Klu in Notch and EE differentiation.** To gain a comprehensive overview of the genes controlled by Klu in the intestine, we performed RNA-Sequencing (RNA-Seq) on FACS-sorted Esg$^+$ progenitor cells expressing either $klu^{RNAi}$ or $UAS-klu^{30}$ (see Fig. 5a, and Methods for details). Principal component analysis on the transcriptome of these populations showed that all sample groups form distinct clusters and that group replicates cluster closely together (Supplementary Fig. 5a). We also

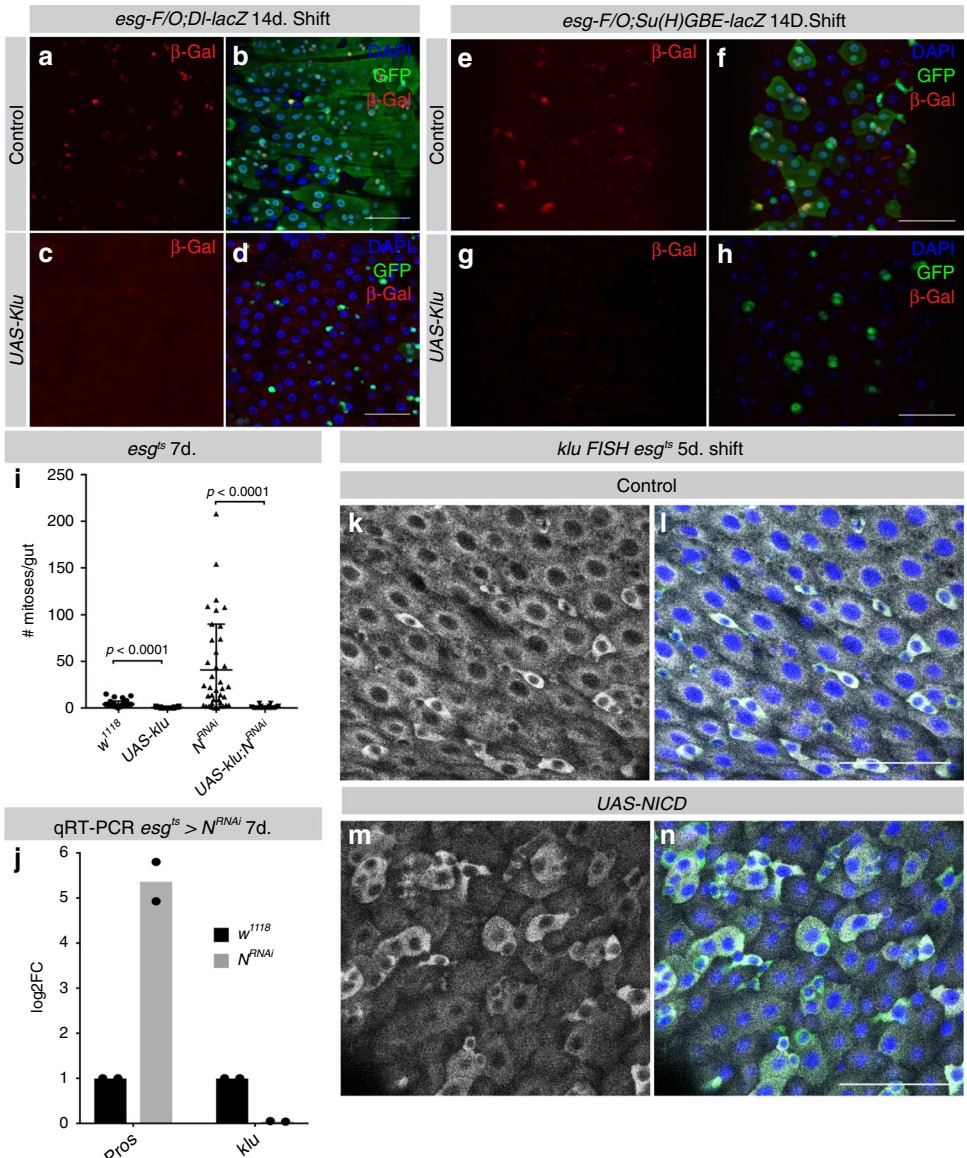

**Fig. 4** Klu is regulated by Notch and represses Notch-induced tumor formation. **a, b** Control *esg-F/O* clones always contain 1 or more ISCs expressing *Dl-lacZ*. **c, d** *esg-F/O>UAS-klu* clones have no detectable *Dl-lacZ* expression. **e, f** Similarly, control *esg-F/O* clones contain EBs expressing *Su(H)-GBE-lacZ*, but not *esg-F/O* clones expressing *UAS-klu* (**g, h**). Representative areas of posterior midgut are shown. $n = 3$ midguts examined/genotype in **a–h**. **i** Quantification of the number of mitoses per midgut in *esg$^{ts}$* guts expressing *N$^{RNAi}$* alone or in combination with *UAS-klu*. Klu expression also reduced mitosis compared to control midguts (*esg$^{ts}$ × w$^{1118}$*, control average = 4.2 mitoses/midgut, compare to *UAS-klu*: 0.48 mitoses/midgut, $P < 0.0001$). $n = 42$ for control (crossed with $w^{1118}$) *esg$^{ts}$* animals and *N$^{RNAi}$* animals, $n = 33$ for *UAS-klu* and $n = 24$ for *UAS-klu;N$^{RNAi}$*. **j** Quantitative real-time PCR of sorted *esg$^{ts}$* GFP$^+$ cells expressing *N$^{RNAi}$* for the EE-marker Pros and Klu. cDNA was derived from two replicates/genotype, each replicate containing mRNA isolated from *esg$^{ts}$* GFP$^+$ cells from 100 midguts/genotype. **k–n** Fluorescent in situ hybridization for a *klu* probe showed induced expression in *esg$^{ts}$* GFP$^+$ cells that overexpress constitutively active Notch intracellular domain (NICD, **m, n**) compared to control *esg$^{ts}$* cells (**k, l**). Representative areas of posterior midgut are shown. $n = 4$ midguts examined/genotype in **k–n**. Scale bar = 50 µm

noticed that the distance between control and *klu$^{RNAi}$* sample groups and the *UAS-klu* group in the largest principal component PC1 is much larger (Supplementary Fig. 5a). This indicates more profound transcriptional changes in the *UAS-klu* samples compared to controls than between *klu$^{RNAi}$* and controls. This is also reflected in the FACS-profile of Esg$^+$ cells expressing *klu$^{RNAi}$* and *UAS-klu*: whereas ISC and EB population sizes appeared similar between control and *klu$^{RNAi}$*, the *UAS-klu*-expressing Esg$^+$ cells showed a loss of clearly distinguishable ISC and EB compartments (Supplementary Fig. 5b–d). We first confirmed that the transcriptome of *esg$^{ts}$>klu$^{RNAi}$* sorted cells indeed reflects the excess EE differentiation phenotype seen in *klu$^{RNAi}$* animals by

performing qRT-PCR for *prospero* (*pros*) and *scute* (*sc*). The EE marker *pros* was upregulated 5-fold upon *klu$^{RNAi}$* (Fig. 5b). The proneural transcription factor Scute (*sc*) is necessary and sufficient for EE generation in the *Drosophila* midgut[14,31,32] and many upstream factors impinge on the expression of *sc* to regulate EE differentiation[33]. mRNA levels of *sc* increased ~2.5-fold upon *klu$^{RNAi}$* expression, and *UAS-klu* expression completely abolished *sc* mRNA expression in stem-progenitor cells (Fig. 5b).

In addition, we checked *klu* mRNA levels to verify knockdown and overexpression efficiency. As expected, we saw a 70% reduction in *klu* mRNA upon *klu$^{RNAi}$*. Surprisingly, however, expression of *klu* mRNA in *UAS-klu*-expressing progenitor cells

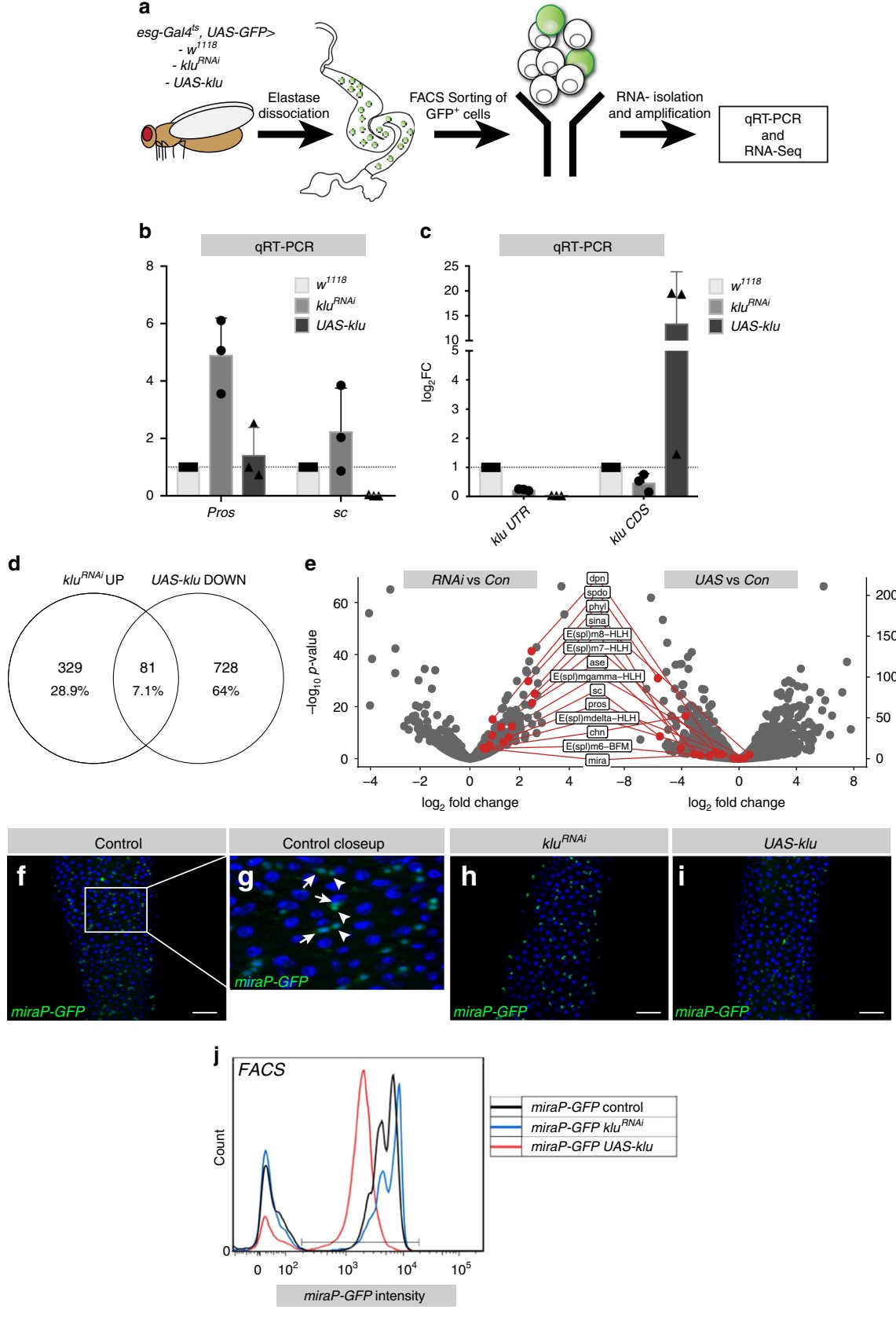

**Fig. 5** RNA-Seq indicates Klu represses Notch targets and EE differentiation genes. **a** Overview of the experiment: $esg^{ts}$ GFP+ cells either expressing $klu^{RNAi}$ or *UAS-kluFL* were sorted in triplicate and their transcriptome was compared to control $esg^{ts}$ GFP cells (see Methods for more details). **b** qRT-PCR analysis of sorted cells for Klu and the critical EE fate regulators Scute (*sc*) and Prospero (*pros*). **c** qRT-PCR analysis of *klu* mRNA expression with a primer pair that targets the endogenous 3′ UTR coding sequence (*klu UTR*) and a primer pair that targets the coding region only (*klu CDS*). For **b** and **c**, cDNA was derived from three replicates/genotype, each replicate containing mRNA isolated from $esg^{ts}$ GFP+ cells from 100 midguts/genotype. Data are plotted as mean ± SEM. **d** Overlap of upregulated genes in $klu^{RNAi}$ and downregulated genes in *UAS-kluFL*. **e** Volcano plots comparing expression of a selection of genes from the overlap of 81 genes shown in **d**. Most genes upregulated in the $klu^{RNAi}$ vs control set (left) are downregulated in the *UAS-klu* vs control set (right). **f–j** Klu represses Da-dependent *miraP-GFP* expression in ISC and EB. **f**, **g** Control *miraP-GFP* expression is high in ISCs (arrowheads) and EBs (arrows). **h** *miraP-GFP* was slightly increased in $klu^{RNAi}$ midguts. **i** *UAS-kluFL* expression resulted in reduced levels of *miraP-GFP*. Representative areas of posterior midgut are shown. $n = 3$ animals examined/genotype in **f–i**. **j** GFP intensity of *miraP-GFP*-positive cells for the genotypes in **f–i** by FACS in a separate experiment. $n = 50$ midguts per genotype. Scale bar = 50 μm

was almost completely abolished (Fig. 5c). This was contrary to the expected *klu* overexpression, but was explained by the fact that the *UAS-klu* construct does not carry the endogenous *klu* 3′ UTR, which our primers targeted. Primers that solely target the coding region of *klu* (*klu CDS*), in turn, readily detected a ~12-fold upregulation of *klu* transcript. Hence, while transgenic Klu was induced as expected, endogenous Klu expression was repressed, indicating that Klu may repress its own expression. This notion of a negative autoregulatory loop was confirmed in our RNA-seq data, as we detected a high number of reads in the coding region of the gene in *UAS-klu* samples, and no reads in the 3′UTR (Supplementary Fig. 6).

Comparing the transcriptomes of wild-type progenitors with the experimental samples, we found 410 genes upregulated in $klu^{RNAi}$ and 809 genes downregulated in *UAS-klu*-expressing Esg+ cells ($P_{adj} < 0.05$, $log_2FC > 0.5$ or $<-0.5$). We also found 283 genes downregulated in $klu^{RNAi}$ and 1025 genes upregulated in *UAS-klu* with the same criteria ($P_{adj} < 0.05$ and $log_2FC < -0.5$ or $>0.5$, Wald significance test with Benjamini and Hochberg correction, see Methods and Supplementary Data 1). Given that only the repressor form (kluZF-ERD) of Klu could recapitulate the phenotype of the expression of full-length Klu in *esg-F/O* clones (Fig. 3), we focused our analysis on genes that would be upregulated in the absence of Klu, but downregulated upon *UAS-klu* expression (Fig. 5d). In this category of 81 genes, many genes involved in the regulation of Notch signaling (the Hairy/Enhancer of Split (*E(Spl)*) complex genes *m6, m7, m8*, and the HES-like transcription factor Deadpan), as well as several previously described regulators of EE differentiation (encoding the proneural proteins Asense (*ase*), Scute (*sc*), and the adaptor protein Phyllopod, *phyl*)) could be identified (Fig. 5e). Additional *E(Spl)* genes (*E(Spl)-mδ* and *E(Spl)-mγ*) were significantly upregulated in $klu^{RNAi}$ samples, but did not change significantly in *UAS-klu* samples (Fig. 5e). *E(Spl)*-genes are a group of genes activated by Notch that mediate its downstream transcriptional response[34]. *Phyl*, in turn, acts to destabilize Tramtrack (*ttk*), a strong repressor of the *achaete–scute* complex genes *scute* and *asense*, loss of which leads to a dramatic increase in EE numbers[33,35]. Reciprocally, loss of *phyl* stabilizes Ttk and results in a complete loss of EEs[36]. The induction of *phyl* in *klu* loss of function conditions thus explains the increase in EEs. We also found that expression of Charlatan (*chn*) was downregulated by *UAS-klu*. Chn is a transcription factor that positively regulates Achaete and Scute, and loss of Chn causes proliferation and differentiation defects in the midgut stem-progenitor compartment[37–39]. Hence, Klu represses the expression of several genes that have reported roles in EE differentiation.

Our transcriptome data also revealed changes downstream of Klu that may explain the Klu-induced exit from the stem cell state: ISC maintenance depends on the Class I bHLH-family member *daughterless* (Da)/E47-like, since loss of Da results in loss of ISC fate and EC differentiation[32]. The gene *miranda* (*mira*) is a Da/

proneural target gene that is also highly expressed in ISCs and to a lesser extent in EBs (Fig. 5f, g)[32,39]. Proneural factors such as Ase and Sc require Da to dimerize and regulate transcription[40]. $klu^{RNAi}$ resulted in a slight but significant upregulation of *mira* in ISC/EB clusters, whereas Klu overexpression resulted in a 2.3-fold downregulation (Fig. 5e). To confirm this, we used a *mira-Promoter-GFP (mira-GFP)* line[32] and combined this with $klu^{RNAi}$ and *UAS-klu*. Confocal microscopy and FACS sorting of cells expressing either $klu^{RNAi}$ and *UAS-klu* confirmed that *UAS-klu* expression could reduce *mira-GFP* levels in Esg+ cells, whereas a slight induction is seen in $klu^{RNAi}$ cells (Fig. 5h, i). FACS sorting indicated an increase in GFP intensity of the EB cells (Fig. 5j, rightmost peak) in $klu^{RNAi}$ Esg+ cells. This suggests that physiologically, Klu acts to inhibit *mira* expression in EBs and that ectopic expression of Klu in ISCs is sufficient to repress the expression of stem cell markers like *miranda*.

**Klu acts upstream of Scute in EE differentiation**. Scute plays a critical role in a transcriptional loop that regulates both ISC proliferation and the initiation of EE differentiation[16]. Our genetic and transcriptional profiling experiments suggest that Klu likely acts downstream of Notch, but upstream of the proneural genes Ase and Sc in repressing EE differentiation (Figs. 3–5, Supplementary Fig. 4). We performed epistasis experiments with Klu and Sc to test this hypothesis. We generated *esg-F/O* clones that express $klu^{RNAi}$ in the presence or absence of $sc^{RNAi}$. Clones expressing $klu^{RNAi}$ contained more EE cells compared to control clones (Fig. 6c, d compare with Fig. 6a, b), whereas clones expressing $sc^{RNAi}$ are almost completely devoid of EE cells (Fig. 6e, f). The combination of $klu^{RNAi}$ and $sc^{RNAi}$ also resulted in clones with little or no EE differentiation (Fig. 6g, h, quantification in Fig. 6i). This suggests that excess EE differentiation in $klu^{RNAi}$-expressing clones depends on Scute. To confirm that Scute would act downstream of Klu in determining EE fate, we combined overexpression of Scute and Klu. Clonal expression of Scute using the *esg-F/O* system resulted in clones consisting almost entirely of Pros-positive EE cells whereas clones expressing *UAS-klu* are completely devoid of EE cells (quantification in Fig. 6j, images in Supplementary Fig. 7a–l). Co-expression of Klu and Scute leads to a marked reduction in clone size (Supplementary Fig. 7m) but EE differentiation was observed in a large fraction of the clones, although the percentage of differentiated cells is reduced compared to *UAS-Sc* alone (Fig. 6j). We conclude that Scute can still induce EE differentiation, even in Klu gain-of-function conditions.

We observed an increase in the number of Pros-pH3 double-positive cells in UAS-Sc compared to control, likely representing the EE-progenitor cells (EEp) undergoing a final round of division[16]. Strikingly, this percentage increased in *esg-F/O > UAS-klu + UAS-sc* clones (Supplementary Fig. 7n). However, the clonal size in this genotype is no larger than in *esg-F/O > UAS-klu*

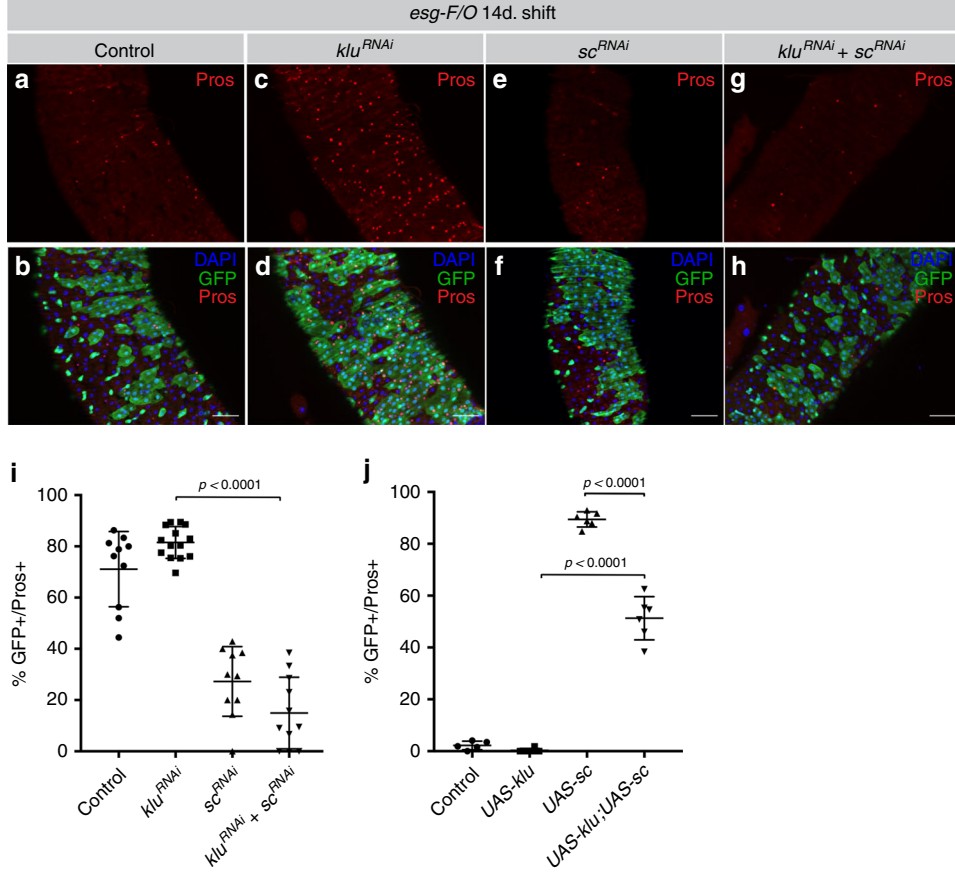

**Fig. 6** Scute acts downstream of Klu in enteroendocrine differentiation. **a**, **b** Control *esg-F/O* clones 14 days after clonal induction. **c**, **d** Expression of *klu*[RNAi] lead to increased EE differentiation in clones, marked by increased numbers of Pros[+]-cells (red) (**c**, **d**, quantification in **i**, **j**). **e**, **f** *sc*[RNAi] clones showed almost no EE differentiation. **g**, **h** Similarly, the combination of *klu*[RNAi] with *sc*[RNAi] resulted in clones lacking EE differentiation. Representative areas of posterior midgut are shown. **i** Quantification of GFP[+]/Pros[+] double-positive cells/clone of the genotypes in **a**–**h**. *n* = 10 for control, *n* = 14 for *klu*[RNAi], *n* = 10 for *sc*[RNAi] and *n* = 12 for *sc*[RNAi];*klu*[RNAi]. **j** Quantification of GFP[+]/Pros[+] double-positive cells/clone of *esg-F/O* clones expressing either *UAS-sc*, *UAS-klu*, or the combination. See Supplementary Fig. 7 for images. *n* = 5 for control, *n* = 6 for *UAS-klu*, *UAS-sc*, and *UAS-klu;UAS-sc*. Error bars represent mean ± SD. Significance was calculated using Student's *t*-test with Welch's correction. Scale bar = 50 μm

over-expressing clones (Supplementary Fig. 7n), indicating that these cells might be arrested in mitosis. This suggests that although Klu expression cannot completely repress *UAS-Sc*-induced EE differentiation, the effect of Klu on cell cycle progression interferes with the proliferation-inducing capacity of Scute.

**Klu binds to genes regulating EE fate, cell cycle and Notch**. The differentially expressed genes (DEGs) from our RNA-Seq analysis might reflect genes that are direct target genes of Klu. Alternatively, the transcriptional changes might be the consequence of a change in cell populations due to the loss or overexpression of Klu. To distinguish between these possibilities and to identify genes directly regulated by Klu, we performed targeted DamID of Klu in Esg[+] stem-progenitor cells[41]. We used the DamID-seq pipeline (ref. [42], see Methods) to identify 1667 genes that had one or more Klu binding peak(s) within 2 kb of their gene body in all three replicates. Using two published position weight matrices for Klu binding[43], we could establish that 692 of the 1667 genes (41.5%) had one or more Klu-binding motif(s) present in their binding peaks. We considered these peaks as high-confidence Klu-bound sites. Our transcriptomics data on Klu indicated that Klu controls many genes involved in Notch signaling, EE differentiation, and cell cycle regulation. We identified a cluster of binding sites at the centrosomal end of the *E(Spl)*-locus around

the *E(Spl)-mδ* and *E(Spl)-mγ* genes (Fig. 7a). Since our RNA-Seq data showed that many of the *E(Spl)*-genes change expression in both *klu*[RNAi] and *UAS-klu* conditions (Fig. 5e), this suggests that Klu could possibly regulate the expression of multiple members of the *E(Spl)*-complex through this binding peak at the centrosomal end of the *E(Spl)*-locus. Furthermore, we identified a Klu-Dam binding peak at the *klu* locus, supporting our hypothesis that Klu acts in an autoregulatory loop by negatively regulating its own expression (Fig. 7b). Previous work has shown that Scute and the *E(Spl)*-complex member *E(Spl)m8-HLH* act in a regulatory loop to generate an EE precursor directly from the ISC[16]. Since our results indicate that Scute is upregulated upon loss of Klu and acts epistatically to Klu in EE formation, we first looked for Klu binding in and around the *scute* locus. We did not observe binding of Klu-Dam around any of the genes in the Achaete/Scute complex. However, we did identify a DamID peak around the *sina* and *sinah* loci (Fig. 7c). Together with the adaptor protein Phyllopod, the Sina and Sinah E3-ubiquitin ligases are able to degrade the transcriptional repressor Tramtrack (*ttk*), which represses EE fate[33,36]. *sina* transcript levels are upregulated 2.2-fold upon *klu* RNAi and *phyl* levels are upregulated 8-fold as well as downregulated 15-fold upon *UAS-klu* expression (Fig. 5e, Supplementary Data File 1). Hence, we propose that Klu represses EE fate determination in EBs upstream of Scute by stabilizing Tramtrack, since Klu directly represses the members of the

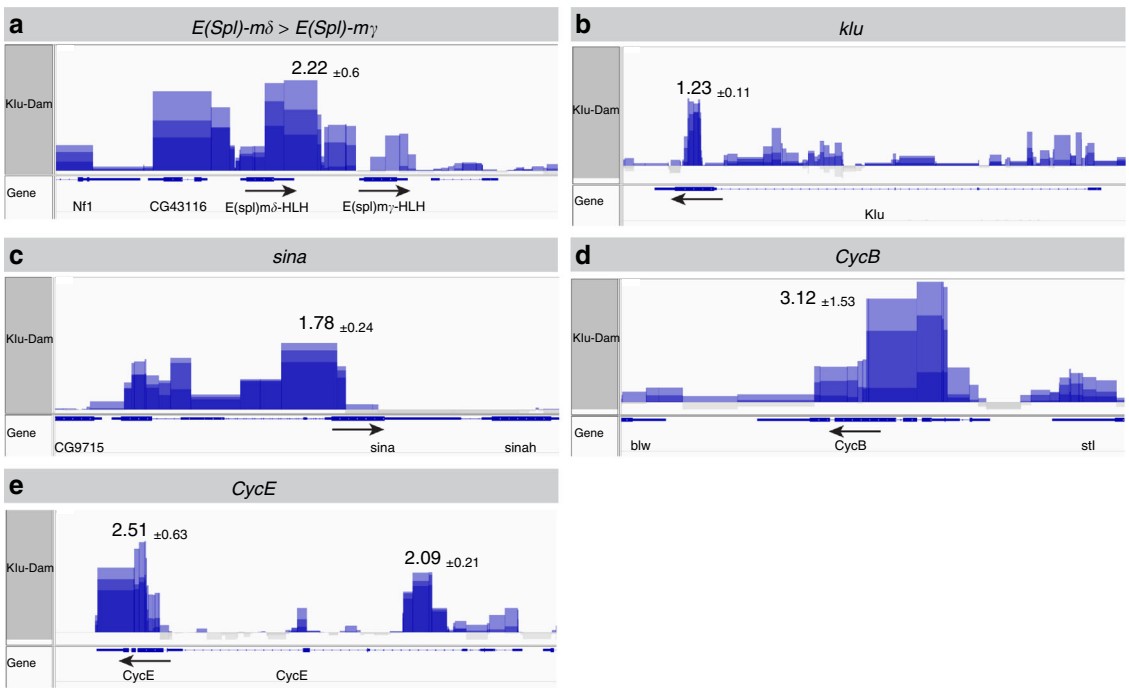

**Fig. 7** Klu binds genes involved in Notch, cell cycle, and EE fate. **a–e** Klu-Dam binding tracks (in triplicate) for the *E(Spl)*-complex locus (**a**), *klu* (**b**), *sina* (**c**), and the Cyclins *CycB* (**d**) and *CycE* (**e**). Tracks are displayed in Integrated Genome Viewer (IGV) as overlayed tracks of the triplicate Klu-Dam vs Dam-only control comparisons. Arrows indicate direction of transcription. Numbers indicate average maximum peak height (log$_2$FC of Klu-Dam over Dam-only control ± SD) for each of the three replicates

E3-ligase complex Sina, Sinah, and (indirectly) Phyl that can normally target Ttk for destruction.

In addition to genes involved in Notch signaling and EE specification, we find evidence for direct repression of critical cell cycle regulators by Klu. We find Klu-binding peaks at both the Cyclin B (*CycB*) and Cyclin E (*CycE*) loci (Fig. 7d, e), two Cyclins that are essential for G1–S and G2–M progression, respectively. CycE is also upregulated upon *klu* RNAi expression. Notch activation is essential for the mitotic-to-endocycle switch in follicle cells of the *Drosophila* ovary, and polyploidization is a critical step in the normal process of EB-to-EC differentiation[44,45]. We propose that Klu plays a role in remodeling the cell cycle in response to Notch activation by directly repressing two critical cell cycle regulators. Furthermore, this explains how Klu acts as a potent suppressor of cell proliferation (Fig. 3, Supplementary Fig. 3).

Altogether, our data suggest a model (Fig. 8) in which Klu acts as a Notch effector in the EB that acts to restrict the duration of the Notch transcriptional response (through negative regulation of the E(Spl)-complex members and Klu itself). Second, Klu prevents activation of the Scute-E(Spl)-m8 transcriptional circuit that triggers EE differentiation. Finally, we find evidence that Klu can bind and repress critical cell cycle regulators such as Cyclin B and Cyclin E, likely promoting the switch from a mitotic to an endoreplicating cell cycle in differentiating ECs.

## Discussion

Our work identified a mechanism by which lineage decisions are cemented through the coordinated repression of alternative fates and of cell proliferation in somatic stem cell daughter cells. Notch-induced expression of Klu in EBs is necessary to repress EE fates in EBs, but also to restrict Notch target gene expression. Hence, its own expression has to be self-regulated to allow differentiation to ECs to proceed. We find that Klu represses several genes that are critical for EE differentiation; most notably genes that influence the

level of Scute. Transient expression of Scute is necessary and sufficient for EE differentiation and this is accomplished by a double-negative feedback loop between Scute and E(Spl)m8 (ref. [16]). Klu expression results in the repression of both transcription factors in EBs, inactivating the transcriptional circuit that governs EE differentiation (Fig. 8). Previous work has shown that Klu is directly regulated by Su(H) and acts as a Notch effector in hemocyte differentiation[46]. We find that overexpression of Klu results in the loss of Notch signaling activity in stem-progenitor cells, and that Klu is able to repress several Notch effector genes (such as the HES/E(Spl) family and HES/E(Spl)-like genes such as Deadpan). We thus propose that Klu acts in a negative feedback loop downstream of Notch signaling to ensure that Notch effector gene activity is transient in EBs, mirroring the transient nature of EE specification by Scute and E(Spl)m8.

Klu is a zinc-finger transcription factor with some similarity to WT1 (refs. [24,47]). While the sequence similarity between these factors is limited, our data suggest that functional parallels can be drawn: Loss of WT1 in the mouse kidney results in glomerulo-sclerosis and is accompanied by ectopic expression of HES/E(Spl) family genes[48] and in zebrafish kidney podocytes Notch expression induces *Wt1* transcription, while the Notch intracellular domain (NICD) and WT1 synergistically promote transcription at the promoter of the HES/E(Spl) family gene Hey1 (ref. [49]). This suggests that the negative feedback between Notch and its effectors Klu or WT1 might be conserved between species, even though conservation at the sequence level between these transcription factors is low.

Our data also support a role for Klu for regulating cell cycle progression. Overexpression of Klu results in a strong block in cell proliferation in $N^{RNAi}$ or oncogenic $Ras^{V12}$-induced tumors and our DamID data suggest that Klu can directly regulate Cyclins B and E. The phenotype of Klu in EBs is in stark contrast to its role in the neuroblast stem cell lineage, where over-expression of Klu leads to a strong overproliferation of immature

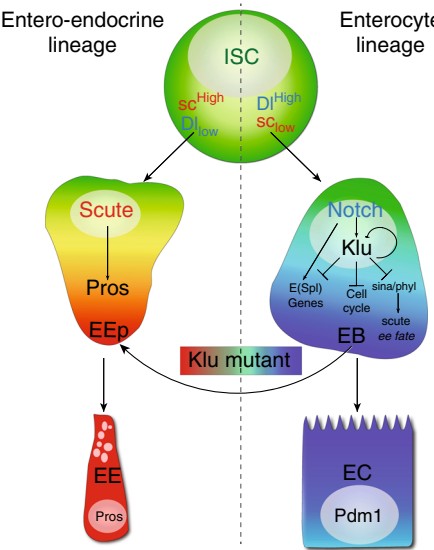

**Fig. 8** Model for Klu function in ISC lineage differentiation. Our data suggest that Klu expression is activated by Delta-Notch signaling in the EB, together with other members of the Hairy/Enhancer of Split family of Notch target genes. Klu accumulation in EBs results in a subsequent repression of these target genes, including the repression of its own expression. Additionally, Klu acts as a safeguard to repress erroneous EE differentiation in the enteroblast–enterocyte lineage by indirectly repressing the accumulation of proneural genes such as Asense and Scute through inhibition of the E3-complex members Phyllopod (phyl) and Seven-in Absentia (sina) that repress the accumulation of Scute and thereby inhibit EE fate. Finally, Klu acts in the regulation of the cell cycle in EB cells, as the cell remodels its cell cycle from a mitotic to an endocycle

neural progenitor cells and the formation of brain tumors[18,19]. However, this likely reflects the different role for Notch in the NB lineage, where continuous activation of Notch similarly leads to INP overproliferation and tumor formation. Thus, the role of Klu in promoting either lineage differentiation or stem-progenitor cell proliferation seems to be context-dependent. Similarly, *Wt1* was initially identified as a tumor-suppressor gene mutated in the rare pediatric kidney cancer Wilms' Tumor[50]. However, expression of WT1 was found to be elevated in many solid tumors and in acute myeloid leukemia[51,52]. During development, WT1 plays a role in the formation of many different tissues of mesodermal and neurectodermal origin[47]. Although WT1 expression seems restricted in adult animals at first glance, whole-body knock-out of WT1 in adult mice results in rapid demise of the animals with kidney, spleen, bone, and fat tissue defects as well as defective erythropoiesis[53]. Furthermore, recent results in zebrafish have shown that *Wt1b* can be re-activated in specific mesenchymal cells upon damage[54], suggesting that *Wt1b* re-expression is involved in regeneration upon damage. In addition, WT1 is often transiently expressed in both nephric and hematopoietic lineages in committed progenitor cell types, similar to the expression of Klu in the EB, raising the possibility that to fully understand the role of WT1-like proteins in tumorigenesis, cell lineage relationships, as well as cell proliferation and differentiation events in tumors need to be taken into account.

Critically, our work highlights the role for transient transcriptional rewiring events during cell specification in stem cell lineages. This rewiring seems to be required to ensure lineage commitment downstream of initial symmetry breaking signals like Notch, and ensure commitment to cell differentiation into a defined lineage. As such, it can be expected that similar

transcriptional rewiring needs to happen for cells to undergo de-differentiation into stem cells in regenerating tissues. Understanding this transcriptional rewiring process will substantially advance efforts to control tissue repair and regeneration in mammals, including humans.

## Methods
**Fly strains and husbandry**. The following strains were obtained from the Bloomington Stock Center: BL28731 (*klu* RNAi on 3rd) BL60469 (*klu* RNAi on 2nd), BL56535 (*UAS-klu[Htol]*), BL11651 (*Dl[05151]-lacZ*) BL26206 (*sc* RNAi), BL51672 (*UAS-sc*), BL1997 (*w[*]; P{w[+ mW.hs] = FRT(w[hs])}2A*), BL4540 (*w[*]; P{w[+ mC] = UAS-FLP.D}JD2*). BL65433 (*y[1] w[*];M{w[+ mC] = hs.min(FRT.STOP1) dam}ZH-51C*) BL1672 (*w[1118]; sna[Sco]/CyO, P{ry[ + t7.2] = en1}wg[en11]*).

VDRC: v27228 (*N* RNAi). Other stocks: *klu-Gal4 UAS-GFP, FRT2A kluR51/Tm6B, hs-Flp, Tub-Gal4, UAS-GFP/Fm7;FRT2A, TubGal80ts/Tm2,Ubx* (T. Klein, Düsseldorf) *UAS-kluFL, UAS-ERD-kluZF, UAS-VP16-kluZF* (all constructs inserted into *ZH51C* on 2nd, C.Y. Lee, U. Michigan) *esg-F/O (w; esg-Gal4, tub-Gal80ts, UAS-GFP; UAS-flp, Act > CD2 > Gal4(UAS-GFP)/TM6B), esg[ts] (y,w;esg-Gal4, UAS-GFP/CyO;tub-Gal80ts/Tm3), Su(H)[ts] (w;Su(H)GBE-Gal4,UAS-CD8-GFP/CyO;tub-Gal80[ts]/TM3), Su(H)GBE-Gal4, UAS-CD8-GFP/CyO;tub-Gal80[ts]/UAS-Flp, Act > CD2 > Gal4, Su(H)-F/O* genotype (control) *w;Su(H)GBE-Gal4, UAS-CD8-GFP/CyO;tub-Gal80[ts]/UAS-Flp, Act > CD2 > Gal4, Su(H)-F/O* genotype (*klu[RNAi]*) *w;Su(H)GBE-Gal4,UAS-CD8-GFP/klu[RNAi BL60469];tub-Gal80ts/UAS-Flp, Act > CD2 > Gal4*, ISC-specific *esg[ts][29] w;esg-GAL4,UAS-2XEYFP/CyO;Su(H)GBE-GAL80,tub-Gal80ts/TM3,Sb, w;esg-gal4, tub-Gal80ts, UAS-GFP/CyO,wg-lacZ;P{w[+ mC] = UAS-FLP.D}JD2/Tm6B*. Stocks generated in this study: *w;If/CyO, P{ry[+ t7.2] = en1}wg[en11];klu[KI]-Gal4/Tm6B* and *w;Klu-Dam(ZH-51C) M4M1/CyO, P{ry[+ t7.2] = en1}wg[en11]*.

**Immunostaining and microscopy**. Midguts were dissected into ice-cold phosphate-buffered saline (PBS), fixed in 4% formaldehyde, and incubated for 1 h at room temperature. Samples were then washed 3 × 10 min, first in 1× PBS with 0.5% Triton X-100, then in 1× PBS with Na-deoxycholate (0.3%), and last in PBT (PBS with 0.3% Triton X-100), and incubated in blocking solution (PBT with 0.5% bovine serum albumin) for 30 min at 4 °C. Samples were incubated with primary antibodies overnight at 4 °C, washed 3 × 20 min at room temperature in PBT, incubated with secondary antibodies diluted in blocking solution at room temperature for 2 h, washed 4 × 20 min with PBT, and mounted in Vecta-Shield (Vector Laboratories)[55]. Antibodies used include Chicken anti-GFP (1:1000; ThermoFisher A10262), mouse anti-Prospero (MR1A, 1:50, DSHB), mouse anti-beta-galactosidase (40-1a, 1:200; DSHB), rabbit anti-beta-galactosidase (1:200; ThermoFisher A11132), mouse anti-Armadillo (N2 7A1, 1:20; DSHB), rabbit anti-phosphorylated Histone H3-Ser10 (pH3S10, 1:500; sc8656-R; Santa Cruz Biotechnology). Images were taken from the R5 and R4 regions of the posterior midgut on a Zeiss Apotome microscope or Zeiss LSM710 confocal at either ×20 or ×40 magnification. Images were captured as Z-stacks with 8–10 slices of 0.22–1.0 μm thickness. Images were converted to maximum-intensity projections in Fiji (https://fiji.sc) and quantifications were performed using the CellCounter FiJi plugin. ROIs in quantifications are defined as images taken from the posterior midgut R4-R5 region at ×20 magnification in which all cells/clones were quantified. Scale bar = 50 μm in all images, except in Fig. 1a: scale bar = 25 μm. Graphing, statistical analysis, and survival curves were produced in GraphPad Prism. Significance was calculated using Student's *t*-test. In case of unequal variances, Student's *t*-test with Welch's correction was used.

**Cloning and transgene generation**. We used the Inducible DamID system from the Van Steensel lab to generate klu-Dam[41]. To this end, we amplified the Klu Full-length cDNA (derived from BDGP Gold clone FI01015) using *AscI* and *NotI*-containing primers and cloned the fragment into the vector *p-attB-min.hsp70P-FRT-STOP#1-FRT-DamMyc[open]* (Addgene plasmid #71809). Transgenic lines were generated by Genetivision Inc. using the phiC31 integrase-mediated site-specific transgenesis system[56]. The finished construct was injected into Bloomington stock BL24482 (*ZH-51C* attP-site on 2nd) and the resulting transgenic lines were tested by genotyping PCR. Both control (Dam-only, BL65433) and klu-Dam transgenic lines were crossed to BL1672 (*w[1118]; sna[Sco]/CyO, P{ry[ + t7.2] = en1}wg[en11]*) before use. The *klu-Gal4KI CRISPR* line was generated by Rainbow Transgenics (Camarillo, CA, USA). A targeting construct was designed to replace the *klu* CDS with the Gal4 CDS at the *klu* ATG. Two independent transformants were obtained that both showed identical EB-specific expression.

**DamID**. Control Dam-only (BL65433) and klu-Dam male flies were crossed to *w; esg-gal4, tub-Gal80ts, UAS-GFP/CyO,wg-lacZ;P{w[ + mC] = UAS-FLP.D}JD2/Tm6B* virgins. Crosses were maintained at 18 °C and progeny was shifted to 29 °C for 24 h to induce the Flp-mediated recombination of the STOP-Cassette. Thirty to 50 midguts of Dam-only and *klu-Dam* were dissected in 1× PBS in three different batches and used for isolation of total genomic DNA. Isolation of methylated GATC-sequences and subsequent amplification was done according to the protocol published by Marshall et al.[57] until Step 34, from which we continued NGS library

preparation using the Illumina TruSeq nano DNA kit LT. After library quality control, samples were sequenced as 50 bp single-end on an Illumina HiSeq2500.

**Midgut FACS RNA isolation and sequencing**. For RNA-Seq, UAS-expression of UAS-klu or klu$^{RNAi}$ was induced using esg-Gal4$^{ts}$, UAS-GFP for 2 days, followed by 16 h of Ecc15 infection to stimulate midgut turnover. We dissected 100 midguts/genotype in triplicate and for each sample 20,000–40,000 cells were sorted into RNAse-free 1× PBS with 5 mM EDTA. RNA was isolated using the Arcturus PicoPure™ RNA Isolation Kit. Subsequently, the entire amount of isolated RNA was used as input for RNA-amplification using the Arcturus™ RiboAmp™ HS PLUS Kit. Two hundred nanograms of amplified aRNA was used as input for RNA-Seq library preparation using the TruSeq Stranded mRNA Library Prep Kit (Illumina) and samples were subsequently sequenced as 50 bp single-end on an Illumina HiSeq2500. For the FACS analysis experiment with DNA staining, we dissected 60–70 midguts/genotype under the same conditions and used NuclearID Red DNA Stain (ENZ-52406, Enzo Life Sciences) for DNA content analysis. FACS-plots were generated with FlowJo v10.

**Quantitative real-time PCR**. Quantitative real-time PCR (qRT-PCR) was performed using amplified RNA from FACS-sorted Esg$^+$ cell populations (see above) as template. cDNA was generated using the QuantiTect Reverse Transcription Kit. qRT-PCR was performed using the TaqMan FAM-MGB system in a 10 μl reaction on a BioRad CFX384 C1000 Touch Cycler using the following probes: klu (dm02361358 s1), pros (dm02135674 g1), sc (dm01841751 s1). Act5C (dm02361909 s1) was used for normalization. The klu CDS primer assay was ordered as a Custom TaqMan Assay. Reactions were performed in triplicate on three independent biological replicates. Relative expression was quantified using the $\Delta\Delta$Ct method. Data were calculated using Microsoft Excel and plotted as relative fold-changes ± SEM in Graphpad Prism.

***klu* FISH**. A 425 bp region in the klu gene, starting at the middle of the 5′UTR and including the first 279 bases of the CDS, was designed to have an Sp6 promoter and a SpeI site at the 5′ end and a downstream T7 promoter and NotI site at the 3′ end. This DNA fragment was synthesized and cloned into a pUCIDT plasmid (IDT). After linearization of the plasmid, transcription and fluorescent labeling of anti-sense and sense klu probes was done following the manufacturer's instruction of the FISH Tag RNA Multicolor kit (Invitrogen Cat. No. MP 32956) using the Sp6 promoter to generate the sense probe and the T7 promoter to generate the anti-sense probe. In situ hybridization was performed using a protocol adapted from the one suggested in FISH Tag RNA Multicolor kit (Invitrogen Cat. No. MP 32956).

**RNA-Seq and DamID data analysis**. The 15–21 million quality-passed reads per sample were mapped to the D. melanogaster reference genome (BDGP6) with TopHat2 (version 2.1.0)[58]. Of each sample, approximately 80% of the reads was mapped to the genome. From this, 90% could be assigned to genes using Fea-tureCounts resulting in 11–15 million analysis-ready reads per sample[59].

The table of raw counts per gene/sample was analyzed with the R package DESeq2 (version 1.16.1) for differential expression[60]. Both sample groups of interest (UAS & RNAi) were pair-wise contrasted with the control sample group (control). For each gene of each comparison, the p-value was calculated using the Wald significance test. Resulting p-values were adjusted for multiple testing with Benjamini & Hochberg correction. Genes with an adjusted p-value <0.05 are considered differentially expressed (DEGs).

For DamID, we used the damid_seq pipeline[42] to generate binding profiles for Klu-Dam. Triplicate samples for Klu-Dam (34.9, 33.5, and 34.1 millions reads) and Dam-only control (34.7, 34.5, and 35.6 million reads) were aligned to the Drosophila genome (UCSC dm6). Overall aligning rate was between 86% and 91% across all samples. First, gat.track.maker.pl script was used to build a GATC fragment file. Then the main utility damidseq_pipeline was used to align the reads to the genome using bowtie2, bin and count reads, normalize counts, and compute log2 ratio between corresponding DamID and control Dam-only samples[42]. The pipeline identified 1707, 1663, 1681 peaks with FDR < 0.01 per each replicate. To test for reproducibly we first used the damid_pipeline[42] to identify peaks with weaker confidence (FDR < 0.1) and the idr python package (https://github.com/nboley/idr) to identify 1169 peaks with IDR < 0.05 between replicate1 and replicate2. We used an in-house developed script to annotate peaks in proximity to genes. In total, 1667 genes found to be in proximity to at least one reproducible peak. To find Klu binding motifs in our reproducible peak set, we scanned for two different Klu PWM (described in ref. [43]) around reproducible peaks using the FIMO tool[61]. Reads were visualized using IGV as overlayed triplicate Klu-Dam (log2FC over Dam-only) tracks.

## Data availability
The authors declare that all data supporting the findings of this study are available within the article and its Supplementary Information files or from the corresponding authors upon reasonable request. DamID data have been deposited in the GEO database under the accession code: GSE131878. RNA-Seq data have been deposited in the GEO database under the accession code: GSE132243.

## Code availability
The in-house developed script to annotate peaks in proximity to genes from damid_pipeline data is available in the file Supplementary Data 2.

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

## Acknowledgements

The authors would like to thank Thomas Klein, C.Y. Lee, Sarah Bray, Claude Desplan, Benoit Biteau, Cai Yu, and Bruce Edgar for providing fly stocks and antibodies, the Bloomington *Drosophila* Stock Center, The VDRC (Vienna) and the Developmental Studies Hybridoma Bank (DSHB) for fly stocks and antibodies, and Kathrin Schubert, Maria Locke, and Karol Szafranski from the Flow Cytometry and Life Science Computing Core Facilities at the FLI-Leibniz Institute on Aging for expert technical assistance. We would also like to thank the FACS and Imaging Facility and Linda Partridge for their support in completing this work at the Max-Planck-Institute for Biology of Aging. This work was supported by DFG research grant number KO5594/1-1 to J.K.

## Author contributions

J.K. and H.J. conceived the project and designed experiments. J.K., S.A., I.A.R.-F., M.B., and E.M. performed experiments and collected data. M.G. helped with optimizing the RNA-Seq library amplification protocol and Dam-ID protocol. T.R.-O. and P.K. performed data analysis on the RNA-Seq and DamID samples. P.S-V. provided preliminary data for the study. J.K. and H.J. wrote the manuscript.

## Additional information

**Competing interests:** The authors declare no competing interests.

