## [Peer Review File · Nature Communications]

Reviewers' Comments:

Reviewer #1:

Remarks to the Author:

This study by Korzelius and colleagues explores the role of Klumpfuss in controlling intestinal stem cell lineage differentiation. Klumpfuss is a transcription factor with a known role in the neuroblast stem cell lineage but no previously described function in the intestine. An enhancer trap line near Klu suggested that there is enteroblast-specific expression of Klu. The authors provide genetic evidence that there is an increase in EE cells upon expression of Klu RNAi in both ISCs and EBs or in a background of Klu heterozygous flies. The overexpression of Klu, in contrast, results in a block of cell division of the ISC. RNAseq analysis of RNA from FACS sorted pooled ISCs+EBs of control background vs Klu RNAi knockdown and Klu overexpression suggested that numerous target genes potentially contribute to the cell cycle effects and EE differentiation phenotypes are found. These data are further supported by Klu DamID identification of Klu binding genome wide in *esg+* ISCs and EBS.

The authors propose a model in which Klu is a target of Notch in the EB and its expression triggers both a cell cycle block and prevention of EE fate in this cell. While this is an elegant model, I feel that the data fall short of providing robust support for it. In particular, the function of Klu in the EB, as opposed to the ISC, or a very indirect non-autonomous function is not entirely clear, nor is its role in directing differentiation.

Major points:

1. An important thesis of this work is that *klu* activity is required in the EB, after Notch signaling has been initiated, to restrict this cell from becoming an EE cell. The authors provide no support for this. They should use the same experimental set-up as in Figure 1G, combined with *klu* RNAi. That is, to specifically express *klu* RNAi in the EB and to lineage-trace the fate of the resulting cells. If their model is correct, these Su(H)G^{BE+} cells should now express EE cell markers. Also, if Klu suppresses cell cycle genes in the EB, then EB knockdown of *klu* should result in cell division in these cells. Is this the case? It is possible that the regulation of EE fate is more complex and less direct than the authors propose. It is also possible that *klu* is required in the ISC to regulate EE activity.
2. Similarly, much of the model presented relies on Klu having a specific expression in the EB which is supported by the Gal4 enhancer trap line of Klu. However, enhancer traps can differ from true protein expression. In fact, in Fly-gut seq, Klu RNA seems more highly expressed in the ISC and in the EC than in the EB. Further support for EB-specific expression should be obtained with an anti-Klu antibody or in situ approaches that could further confirm that Klu specific expression?
3. Page 9, figure 3- they talk about impaired differentiation but mention nothing on division. Since there are no clones in B and D, isn't this more suggestive of the fact that ISCs are not dividing and that differentiation is impaired because the cells are not dividing? The authors should include PHH3 and DI staining and quantitation here. It seems that a more likely conclusion is that constitutive *klu* expression inhibits stem cell division.
4. For figure 2G, the authors state that the reason there is no difference between GFP positive and GFP negative (inside and outside MARCM clone) is because "in this genotype the GFP negative tissue is heterozygous for *klu* mutant" but they do not expand on this further. Are they saying that *klu* is haploinsufficient and that heterozygous mutants will result in *klu* phenotype of excess EEs? How can they rule out that the homozygous mutants do not have a cell non-autonomous effect causing other cells to exhibit mutant phenotype? Also, how could they rule out that other factors in the genetic background of *klu* heterozygotes are contributing to this mutant phenotype? MARCM clones with the RNAi line should be done as well as the above suggestion regarding knockdown

with lineage tracing in the EB.

5. The authors conclude on p 10 that "Altogether, our results indicate that the N-mediated transient expression of Klu in EBS is critical to restrict lineage commitment to the EC fate and inhibit proliferation, but that normal differentiation can only proceed once Klu is downregulated". I find this an over-interpretation of the data presented. As far as I can tell, they provide no data suggesting that Klu expression in EBs is Notch dependent. I am also not convinced that they provide evidence that the physiological downregulation of Klu proceeds and is necessary for lineage restriction and inhibition of proliferation.

6. Similarly, on p7 they state that "Together with the temporarily restricted endogenous expression of klu"- where is the evidence that it is "temporarily" restricted? If they are referring to EB-specific expression, they should provide additional support as mentioned in point 1.

7. With regards to the epistasis experiments where they show that klu acts downstream of Notch in Notch RNAi induced tumor formation in the intestine- How could they rule out that it could be acting in parallel? Same for RasV12. Blocking cell division in any way would give the same epistasis results.

8. If the proposed model is correct, the overactivation of Notch in a klu mutant context should result in defects of EC differentiation and EE fate acquisition. Is this true?

9. With regards to the cell sorting for the transcriptomics, why did the authors sort for esg? It is possible that after 2 days of klu RNAi expression or UAS klu, there would be a shift in ISC and EB populations so that in the Klu overexpression it would be only an ISC population and in the klu RNAi there may be more EBs? It is difficult to know whether the differences at the RNA level are due to potential shifts in the populations. They should provide evidence of the cell populations present upon 2 days of expression in these conditions.

10. Further support of Klu regulating downstream targets regulating EE targets would be beneficial to the manuscript. For example, using available antibodies to look in EBs expressing klu RNAi.

11. The authors should provide full datasets of the RNAseq and the damID data. In Figure 6, are the peaks that are shown called by the peak calling pipeline (ie, are they significant?).

12. Figure 2E presents the quantification of no. of pros+ cells/ROI without mentioning what specifies their "ROI".

13. Figure 1H presents the quantification of "%GFP+/Pros+" but what is being quantified isn't clear.

14. Figure 3E states that n=5, what is n here? Guts or ROI?

Minor

- For figure 1 F'- It would be helpful if the GFP signal can be increased to better see clonal GFP.
- Quantification of S2 E is not in the same order as the figure.
- In the text on page 17, instead of referring to figure 6G, it is meant to be figure 6F.
- Figure 4 D- text reads 410 upregulated genes upon klu RNAi instead of 329 on figure and 809 instead 728 for genes downregulated upon klu overexpression.

Reviewer #2:

Remarks to the Author:

This is a high quality paper which illustrates nicely the power of *Drosophila* as a system to reveal pathways that regulate key developmental processes.

The authors show convincingly that the zinc finger transcription factor Klumpfuss (Klu) acts downstream of notch and upstream of Scute to restrict the fate of enteroblasts by suppressing the enteroendocrine cell fate. A sophisticated use of loss and gain of function coupled with epistatic analysis supports these conclusions. Mechanistically transcriptomics coupled with DAMID profiling support a model by which Klu directly represses E(Spl)m8-HLH and Phyllopod which in turn negatively regulate Scute. Furthermore, Klu suppresses cell cycle genes, committing EBs to differentiation.

I am supportive of publication but would like the authors to respond to a couple of issues. Firstly, the authors refer to 4 published papers when stating Klumpfuss is likely to be a transcriptional repressor. However, as far as I can see, these papers do not show whether Klumpfuss is a repressor or activator but speculate on both possibilities. If one is to take the analogy with WT1 seriously (see below), WT1 is an activator and repressor but far more target genes have been shown to be activated than repressed.

To investigate this possibility the authors overexpressed wildtype Klu or Klu fused to a strong activator or repressor domain in ISC derived clones. Even the wildtype protein when ectopically expressed not only inhibited EE differentiation but also EC differentiation, reflecting the toxic nature of this protein. The repressor fusion protein recapitulated this phenotype, whereas the activation construct did not. Ideally, these fusion constructs should be inserted into the endogenous locus and their effects determined. Notwithstanding this gripe, these experiments do support the idea Klu is acting mainly as a repressor. The authors look for repressed targets by documenting mRNAs whose expression goes up when Klu is knocked down but decreased when Klu is overexpressed. There are 81 such genes. Direct targets for repression are then identified using DAMID. This is an impressive approach which does reveal key downstream targets. However, from the transcriptomic data it seems there are a number of mRNAs whose levels decrease in Klu loss of function and a set that increased in the gain of function context. Are there any mRNAs that decrease in the loss while increasing in the gain context. In other words potentially activated by Klu. All I ask is that the authors investigate this issue and whether any such genes are direct targets in the DAMID analysis. I would not expect further experimentation.

My second point concerns the designation "WT1-like" in the title and referred to superficially in the discussion. The paper is certainly interesting enough without including WT1 in the title. However if this helps to increase the readership it could stay in. The question is why "WT1-like"? I suppose because Klu has 4 *egr*-like zinc fingers, just like WT1. Is there any other reason to say WT1 rather than just another member of the family? At least the authors don't suggest Klu is a WT1 orthologue as suggested by others. WT1 has an unusual first zinc finger, which is implicated in RNA binding. Does Klu have a similar first zinc finger? Is there any homology outside the zinc fingers? What about any possible conserved developmental functions? WT1 functions in the mesoderm and neural ectoderm but there is no evidence it plays a role in the intestine or endoderm. However, WT1 does play a key role in sensory neuron development as does Klu in *Drosophila*. WT1 plays complex roles in the balance between mesenchymal and epithelial states. If the authors want to keep WT1 in the title perhaps they should provide some more discussion along the lines discussed here. There are very recent comprehensive reviews on WT1, e.g Hastie, *Development* 2017. I suppose the key issue is whether a possible relationship to WT1 adds insight into either the role of WT1 in various processes and disease or the role of Klu in flies.

Response to reviewers NCOMMS-18-13852-T

We would like to thank the reviewers for the positive reception of our work. The reviewers agree that our study is of significant interest and that the findings are reported in a 'high quality paper' (reviewer 2) and result in an 'elegant model' (reviewer 1). A number of concerns were noted, however, and additional insight was requested by reviewer 1 to support our model. We have now addressed these concerns with additional experiments and by editing the manuscript.

The main additions to our original data are the following:

- *We have performed a Su(H)-FlipOut > klu^{RNAi} lineage-tracing experiment and find that loss of Klu in EBs can result in a partial fate transformation from EB to EE fate.*
- *We have generated a Gal4 knock-in line replacing the klu CDS with the Gal4 CDS at the ATG-start codon of Klu to further confirm the expression of Klu in EBs.*
- *We have added fluorescent in situ hybridization (FISH) data for Klu, confirming klu mRNA expression in the EBs.*
- *We have added MARCM clones expressing klu RNAi to exclude non-autonomous effects of Klu repression on EE differentiation*
- *We have added qRT-PCR data from *esg^{ts}* > *N^{RNAi}* tumors, showing a complete loss of klu expression from the Esg-positive ISC-EB compartment. This supports the notion that Notch is necessary for the expression of Klu in the stem-progenitor compartment.*

In the following, we list our responses to the reviewers' concerns in detail:

Reviewer #1 (Remarks to the Author):

This study by Korzelius and colleagues explores the role of Klumpfuss in controlling intestinal stem cell lineage differentiation. Klumpfuss is a transcription factor with a known role in the neuroblast stem cell lineage but no previously described function in the intestine. An enhancer trap line near Klu suggested that there is enteroblast-specific expression of Klu. The authors provide genetic evidence that there is an increase in EE cells upon expression of Klu RNAi in both ISCs and EBs or in a background of Klu heterozygous flies. The overexpress of Klu, in contrast, results in a block of cell division of the ISC. RNAseq analysis of RNA from FACS sorted pooled ISCs+EBs of control background vs Klu RNAi knockdown and Klu overexpression suggested that numerous target genes potentially contribute to the cell cycle effects and EE differentiation phenotypes are found. These data are further supported by Klu DamID identification of Klu binding genome wide in *esg+* ISCs and EBS.

The authors propose a model in which Klu is a target of Notch in the EB and its expression triggers both a cell cycle block and prevention of EE fate in this cell. While this is an elegant model, I feel that the data fall short of providing robust support for it. In particular, the function of Klu in the EB, as opposed to the ISC, or a very indirect non-autonomous function is not entirely clear, nor is its role in directing differentiation.

Major points:

1. An important thesis of this work is that klu activity is required in the EB, after Notch signaling has been initiated, to restrict this cell from becoming an EE cell. The authors provide no support for this. They should use the same experimental set-up

as in Figure 1G, combined with *klu* RNAi. That is, to specifically express *klu* RNAi in the EB and to lineage-trace the fate of the resulting cells. If their model is correct, these Su(H)GBE+ cells should now express EE cell markers. Also, if *Klu* suppresses cell cycle genes in the EB, then EB knockdown of *klu* should result in cell division in these cells. Is this the case? It is possible that the regulation of EE fate is more complex and less direct than the authors propose. It is also possible that *klu* is required in the ISC to regulate EE activity.

The thesis that Klu activity is required in the EB is indeed central to our model and the suggested lineage-tracing experiment provides additional proof for this. We have performed the requested Su(H)-FlipOut > kluRNAi lineage-tracing experiment and added the images and quantification to Figure 2 of the Main Text. The results clearly show that loss of Klu in EBs can result in a partial fate transformation in EBs from EC to EE fate.

2. Similarly, much of the model presented relies on *Klu* having a specific expression in the EB which is supported by the Gal4 enhancer trap line of *Klu*. However, enhancer traps can differ from true protein expression. In fact, in Fly-gut seq, *Klu* RNA seems more highly expressed in the ISC and in the EC than in the EB. Further support for EB-specific expression should be obtained with an anti-*Klu* antibody or in situ approaches that could further confirm that *Klu* specific expression?

We concur that our expression data being solely derived from a single klu-Gal4 reporter line should be supported by another independent reporter approach. Our original klu-Gal4 line was derived from the original paper describing the klumpfuss mutant by Klein and Campos-Ortega (1997, Development) and the Gal4-insertion is located upstream at the 5'end of the klu coding sequence. We have now generated a Gal4 knock-in line replacing the klu CDS with the Gal4 CDS at the ATG-start codon of Klu to rule out any effects of an upstream Gal4 insertion on transcription. 2 independent lines show the exact same pattern of expression for Klu as the original klu-Gal4 reporter line. We have added images of the Klu-Gal4 CRISPR line in combination with UAS-GFP and DI-lacZ and Su(H)GBE-lacZ markers as Figure S1 of our revised manuscript. Furthermore, we have performed fluorescent in situ hybridization on Su(H)GBE-Gal4 > UAS-GFP animals that also showed klu mRNA expression predominantly in the EBs.

3. Page 9, figure 3- they talk about impaired differentiation but mention nothing on division. Since there are no clones in B and D, isn't this more suggestive of the fact that ISCs are not dividing and that differentiation is impaired because the cells are not dividing? The authors should include PHH3 and DI staining and quantitation here. It seems that a more likely conclusion is that constitutive *klu* expression inhibits stem cell division.

We agree with the reviewer that UAS-klu-expressing clones show impaired proliferative potential in addition to impaired differentiation compared to wild-type clones. We have adjusted the main text to include a statement about Klu expression impairing clonal proliferation.

However, we note that it is unlikely that Klu inhibits differentiation primarily by inhibiting ISC cell cycle progression: First, our damID experiments show that Klu directly binds E(Spl) complex as well as sina / sinah genes. Second, previous work from the Basto and Bardin labs has shown that impairing cell cycle progression in ISCs does not prevent differentiation, but rather results in premature differentiation of ISCs (Gogondeau et al., Nature Comms. 2015). Hence, ISC differentiation seems to be the default mode upon cell cycle inhibition in ISCs. We have confirmed this in unpublished data perturbing other critical mitotic regulators besides Bub3, which

results in similar premature differentiation into an EC-like state (Khaminets and Jasper, unpublished). Further studies are required to understand the complex interplay between cell cycle progression and differentiation in this stem cell lineage.

4. For figure 2G, the authors state that the reason there is no difference between GFP positive and GFP negative (inside and outside MARCM clone) is because “in this genotype the GFP negative tissue is heterozygous for *klu* mutant” but they do not expand on this further. Are they saying that *klu* is haploinsufficient and that heterozygous mutants will result in *klu* phenotype of excess EEs? How can they rule out that the homozygous mutants do not have a cell non-autonomous effect causing other cells to exhibit mutant phenotype? Also, how could they rule out that other factors in the genetic background of *klu* heterozygotes are contributing to this mutant phenotype? MARCM clones with the RNAi line should be done as well as the above suggestion regarding knockdown with lineage tracing in the EB.

To rule out any non-autonomous effects of klu^{R51} heterozygosity, we have added MARCM clones expressing klu^{RNAi} and added these data to Supplementary Figure S2. We find no evidence for a cell-non-autonomous role for Klu in EE differentiation, measured by quantifying EE numbers inside and outside the clonal area in control and klu^{RNAi} clones. We have changed the main text to clarify this issue.

5. The authors conclude on p 10 that “Altogether, our results indicate that the N-mediated transient expression of Klu in EBS is critical to restrict lineage commitment to the EC fate and inhibit proliferation, but that normal differentiation can only proceed once Klu is downregulated”. I find this an over-interpretation of the data presented. As far as I can tell, they provide no data suggesting that Klu expression in EBs is Notch dependent. I am also not convinced that they provide evidence that the physiological downregulation of Klu proceeds and is necessary for lineage restriction and inhibition of proliferation.

*To show that Klu expression is dependent on Notch, we have added qRT-PCR data from $esg^{ts} > N^{RNAi}$ cells. When expressing N^{RNAi} , we see a complete loss of *klu* expression from the *Esg*-positive ISC-EB compartment. This supports the notion that Notch is necessary for the expression of Klu in the stem-progenitor compartment. This is consistent with the results from Terriente-Felix et al., (2013, Development) showing that Klu expression can be directly regulated by the Notch effector Su(H).*

6. Similarly, on p7 they state that “Together with the temporarily restricted endogenous expression of *klu*”- where is the evidence that it is “temporarily” restricted? If they are referring to EB-specific expression, they should provide additional support as mentioned in point 1.

*We have re-phrased this in the main text to clarify this issue: Our expression data demonstrate that Klu is expressed exclusively in EB cells and not in ISCs (See *klu-Gal4* CRISPR-data in Figure 1). Furthermore, we argue that the EB is inherently a transient state, defined by high Notch activity (*Su(H)GBE-lacZ*), in between the differentiation from ISC to EC. Differentiating enterocytes no longer show expression of this Notch reporter, suggesting that Notch activity needs to be activated and subsequently turned off for differentiation to proceed. We therefore argue that Notch activation and Klu expression are transient events and the EB a transient state between ISC and EC. However, we agree that we did not formally demonstrate ‘temporally restricted’ expression of Klu and have therefore deleted references to the transient or temporally restricted nature of Klu expression.*

7. With regards to the epistasis experiments where they show that *klu* acts downstream of Notch in Notch RNAi induced tumor formation in the intestine- How could they rule out that it could be acting in parallel? Same for RasV12. Blocking cell division in any way would give the same epistasis results.

We agree with the Reviewer that based only on the results in Figure S2, one can conclude that Klu could also act in parallel to suppress tumor growth of Notch RNAi tumors, since UAS-klu expression inhibits proliferation. However, the new data we present in Supplementary Figure S3F show that there is no Klu expression in $esg^{ts} > N^{RNAi}$ tumor cells, demonstrating a requirement for Notch in Klu expression. The experiments in Figure S3A-E show that re-expressing Klu in a Notch RNAi context is sufficient to repress tumor formation. These results, together with data from Terriente-Felix et al. (2013), strengthen our hypothesis that Klu is activated by Notch and that Klu activation is necessary for EC differentiation and inhibition of the mitotic cell cycle. In the main text, we conclude that "This suggests that Klu acts downstream of Notch in EC differentiation, but additionally acts as a potent inhibitor of cell proliferation". This is highlighted by our experiments with RasV12 overexpression, a potent inducer of cell division. These demonstrate that Klu overexpression acts as an inhibitor on cell proliferation, but we do not wish to propose that Klu acts in the same genetic pathway as EGFR/Ras signaling in controlling ISC proliferation. We have clarified this in the main text to avoid any confusion.

8. If the proposed model is correct, the overactivation of Notch in a *klu* mutant context should result in defects of EC differentiation and EE fate acquisition. Is this true?

*We have expressed the constitutively active intracellular domain of Notch (Notch ICD or N^{intra}) with or without klu^{RNAi} in the stem-progenitor compartment using esg^{ts} , UAS-GFP. Ectopic activation of Notch leads to a rapid loss of the Esg^{+} -compartment through differentiation into ECs (Figure S3G-J). Concomitant knockdown of Klu does not affect this phenotype. This would suggest that Klu would not act downstream of Notch, but rather in parallel to Notch in EB > EC differentiation. Alternatively, strong Notch activation could act to rapidly induce differentiation before full knockdown of Klu is achieved. Indeed, after 3 days of induction, the majority of guts (6/7 guts) show widespread loss of the Esg^{+} -compartment in $esg^{ts} > UAS-N^{intra}$ animals. This is at a timepoint at which we normally do not see a significant change in EE differentiation rate in $esg^{ts} > klu^{RNAi}$ animals. We speculate that strong Notch activation can mask the *klu* LOF phenotype in EC/EE differentiation. In addition, Notch activation triggers multiple other target genes that might act redundantly to affect EC/EE differentiation.*

9. With regards to the cell sorting for the transcriptomics, why did the authors sort for *esg*? It is possible that after 2 days of *klu* RNAi expression or UAS *klu*, there would be a shift in ISC and EB populations so that in the Klu overexpression it would be only an ISC population and in the *klu* RNAi there may be more EBs? It is difficult to know whether the differences at the RNA level are due to potential shifts in the populations. They should provide evidence of the cell populations present upon 2 days of expression in these conditions.

*We have sorted for *esg*-Gal4-driven GFP since *Esg* is strongly expressed in both ISCs and EBs. To address a possible shift in population sizes of ISCs and EBs, we performed a FACS-analysis experiments using the same genotypes and timepoints as used for RNA-Seq of *Esg*+ populations (Figure S5). Based on DNA content and GFP-intensity, control and klu^{RNAi} Esg^{+} populations appear very similar, with clearly distinguishable ISC and EB compartments. Interestingly, the expression of UAS-*klu**

leads to a loss of the clear distinction between ISC and EB populations in the *Esg*⁺-compartment (Figure S5B-D). This is in line with our results with *DI-lacZ* and *Su(H)GBE-lacZ* reporter gene expression upon *UAS-klu* expression in *esg-F/O* clones (Figure 3), which suggests the loss of normal *DI-Notch* signaling between *ISC-EB* pairs. Although there was no clear difference control and *klu*^{RNAi} *Esg*⁺ populations, our RNA-Seq data showed upregulation of several key *EE* fate regulators (*Pros*, *Scute*, *Asense*), reflecting a shift towards entero-endocrine cell fate in *Esg*⁺ cells expressing *klu*^{RNAi}. Hence, the goal of the RNA-Seq experiment to capture early changes happening in the stem-progenitor compartment (*ISC* and *EB*) upon loss of *Klu* was successful. To further distinguish indirect (i.e. by shift in populations) and direct targets of *Klu*, we resorted to *DamID* of *Klu* as an additional criterium for direct regulation by *Klu*.

10. Further support of *Klu* regulating downstream targets regulating *EE* targets would be beneficial to the manuscript. For example, using available antibodies to look in *EBs* expressing *klu* RNAi.

We agree that it is important to show that EE genes are regulated by Klu, and, accordingly, we have shown an interaction between N, Klu, and Scute, which places Klu upstream of the regulatory mechanism that determines EE fate (Fig. 5). We have further included an analysis of Miranda expression in Klu gain- and loss-of-function conditions, which validates our transcriptome analysis (Fig. 4F-I).

11. The authors should provide full datasets of the RNAseq and the *damID* data. In Figure 6, are the peaks that are shown called by the peak calling pipeline (ie, are they significant?).

We will provide full access to both the DamID and RNA-Seq data. DeSeq2 results for the RNA-Seq data are provided as a table (Sup. Table S1). Raw Sequencing data (RNAseq and DamID) will be deposited to the European Nucleotide Archive (ENA) at the time of acceptance. All peaks shown in Figure 6 are from significant hits from the DamIDseq pipeline. We have included a statement on this in the Figure legend for Figure 6.

12. Figure 2E presents the quantification of no. of *pros*⁺ cells/ROI without mentioning what specifies their "ROI".

Region of interest (ROI) here is the imaged posterior midgut region (regions R4+R5). ROI's were derived from 6-9 different intestines/animals per condition. We added this statement to the figure legend and also added the number of animals for each condition.

13. Figure 1H presents the quantification of "%GFP+/Pros+" but what is being quantified isn't clear.

In Figure 1H we quantified the number of GFP+Pros double-positive cells per ROI. ROI here is the imaged posterior midgut region (regions R4+R5) at 20X magnification. We have clarified this in the figure legend.

14. Figure 3E states that n=5, what is n here? Guts or ROI?

n is 5 intestines/animals per condition: for each animal, 1 ROI was taken from the posterior midgut (R4-R5 region) and the total number of GFP-Pros double-positive cells was counted. We have clarified this in the Figure legend.

Minor

- For figure 1 F'- It would be helpful if the GFP signal can be increased to better see clonal GFP.

We have adapted this panel in our revised manuscript.

- Quantification of S2 E is not in the same order as the figure.

We have adapted this panel in our revised manuscript.

- In the text on page 17, instead of referring to figure 6G, it is meant to be figure 6F.

We have corrected this error in our revised manuscript.

- Figure 4 D- text reads 410 upregulated genes upon klu RNAi instead of 329 on figure and 809 instead 728 for genes downregulated upon klu overexpression.

The total number of genes for each dataset is 410 and 809 respectively. In the Venn diagram in Figure 4E the 81 genes in the overlap are present in both datasets: e.g. for the klu^{RNAi} UP: $329 + 81 = 410$. We have changed the Main Text to clarify this further ("In this category of 81 genes present in both datasets").

Reviewer #2 (Remarks to the Author):

This is a high quality paper which illustrates nicely the power of *Drosophila* as a system to reveal pathways that regulate key developmental processes.

The authors show convincingly that the zinc finger transcription factor Klumpfuss (Klu) acts downstream of notch and upstream of Scute to restrict the fate of enteroblasts by suppressing the enteroendocrine cell fate. A sophisticated use of loss and gain of function coupled with epistatic analysis supports these conclusions. Mechanistically transcriptomics coupled with DAMID profiling support a model by which Klu directly represses E(Spl)m8-HLH and Phyllopod which in turn negatively regulate Scute. Furthermore, Klu suppresses cell cycle genes, committing EBs to differentiation.

I am supportive of publication but would like the authors to respond to a couple of issues.

Firstly, the authors refer to 4 published papers when stating Klumpfuss is likely to be a transcriptional repressor. However, as far as I can see, these papers do not show whether Klumpfuss is a repressor or activator but speculate on both possibilities. If one is to take the analogy with WT1 seriously (see below), WT1 is an activator and repressor but far more target genes have been shown to be activated than repressed.

To investigate this possibility the authors overexpressed wildtype Klu or Klu fused to a strong activator or repressor domain in ISC derived clones. Even the wildtype protein when ectopically expressed not only inhibited EE differentiation but also EC differentiation, reflecting the toxic nature of this protein. The repressor fusion protein recapitulated this phenotype, whereas the activation construct did not. Ideally, these fusion constructs should be inserted into the endogenous locus and their effects determined. Notwithstanding this gripe, these experiments do support the idea Klu is acting mainly as a repressor. The authors look for repressed targets by documenting mRNAs whose expression goes up when Klu is knocked down but decreased when Klu is overexpressed. There are 81 such genes. Direct targets for repression are then identified using DAMID. This is an impressive approach which does reveal key downstream targets. However, from the transcriptomic data it seems there are a number of mRNAs whose levels decrease in Klu loss of function and a set that increased in the gain of function context. Are there any mRNAs that decrease in the loss while increasing in the gain context. In other words potentially activated by Klu. All I ask is that the authors investigate this issue and whether any such genes are direct targets in the DAMID analysis. I would not expect further experimentation.

*We would like to thank the reviewer for the positive reception of our work, and we agree that transcription factors rarely act as sole inhibitors or activators of transcription: this depends on context and co-factors at the respective target promoters. To address the concern, we have added a table with our RNA-Seq results for all categories (Up- and Downregulated for both *klu*^{RNAi} and UAS-*klu*) to the revised manuscript and we will also submit raw read-data to the European Nucleotide Archive (ENA) at the time of acceptance. There are indeed genes that are downregulated upon *klu*RNAi and upregulated upon UAS-*klu* expression, but we have focused our analysis on the genes that are likely repressed by Klu based on our analysis of the UAS-*klu*ZF-ERD FlipOut clones (Figure 3). We have clarified this point in the Main Text of our revised manuscript.*

With respect to the potential different levels of expression for the independent transgenes: the fusion constructs that we used for Figure 3 were all inserted into the same genomic locus (ZH51C) on the 2nd Chromosome by attP-mediated site-directed insertion (Janssens et al., 2017 Dev. Cell), making any difference in expression between these different fusion constructs unlikely. We have added this information to the Results and the Methods section of our revised manuscript).

My second point concerns the designation “WT1-like” in the title and referred to superficially in the discussion. The paper is certainly interesting enough without including WT1 in the title. However if this helps to increase the readership it could stay in. The question is why “WT1-like”? I suppose because Klu has 4 egr-like zinc fingers, just like WT1. Is there any other reason to say WT1 rather than just another member of the family? At least the authors don't suggest Klu is a WT1 orthologue as suggested by others. WT1 has an unusual first zinc finger, which is implicated in RNA binding. Does Klu have a similar first zinc finger? Is there any homology outside the zinc fingers? What about any possible conserved developmental functions? WT1 functions in the mesoderm and neural ectoderm but there is no evidence it plays a role in the intestine or endoderm. However, WT1 does play a key role in sensory neuron development as does Klu in *Drosophila*. WT1 plays complex roles in the balance between mesenchymal and epithelial states. If the authors want to keep WT1 in the title perhaps they should provide some more discussion along the lines discussed here. There are very recent comprehensive reviews on WT1, e.g. Hastie, *Development* 2017. I suppose the key issue is whether a possible relationship to WT1 adds insight into either the role of WT1 in various processes and disease or the role of Klu in flies.

We have added a more in-depth discussion of the parallels between WT1 and Klu in our updated discussion and cited the suggested review, which gives a good overview of the functions of WT1 in mammalian systems. We hope to explore further parallels between WT1 and Klu in lineage determination in mammalian adult stem cell lineages in more detail in the future.

Reviewers' Comments:

Reviewer #1:

Remarks to the Author:

The authors have satisfactorily addressed a number of my points (1, 2, 4, 5, 9, 10, 12, 13, 14, and my minor points).

I still feel that there are two logical flaws in the interpretation of the data presented in this manuscript. The first has to do with the interpretation of the role of Klu overexpression in differentiation. If cell division is blocked, it follows that cell differentiation cannot occur. I

-I do not feel that the authors addressed my point 3 regarding this. They still maintain throughout their entire section on p9 that this is affecting "differentiation".

I do not believe that they "adjusted the main text to include a statement about Klu expression impairing clonal proliferation". They also did not add the PHH3 and DI staining and quantification that I suggested. Certainly, their unpublished data on a link with cell cycle and differentiation is interesting, but it does not impact the interpretations made in this manuscript.

-I still maintain that the authors cannot conclude on p9 that overexpression of Klu inhibits stem cell differentiation without at least saying in the same sentence that it blocks cell division. Of course, if there is no cell division, there cannot be any differentiation. The focus of the paragraph is on differentiation and division is not mentioned.

Presumably, the inhibition of cell division upon the overexpression of Klu is due to activity in the ISC (where it is not normally expressed). The physiological relevance of this to me is not clear.

This being said, I do appreciate that their new data on Su(H)GBE-F/O klu RNAi do provide supporting evidence for an important role in Su(H)GBE+EBs to restrict EE fate. I would caution the authors to be careful in overinterpreting data.

The second problem that I see with the logic is regarding their interpretation of the role of Klu overexpression in Notch signaling: obviously if Delta expression is blocked, no downstream signaling can occur. I would say that this is one effect and not two, as they maintain "This suggests that Klu expression promotes an exit from the (DI+) stem cell state, but also interferes with transcriptional programs induced by Delta-Notch signaling in EBs." In addition, the loss of DI expression upon overexpression is likely occurring in the ISC where Klu is not normally expressed and again, how this relates to the normal physiological context where Klu is only expressed in EBs is not clear to me.

-I am not sure to understand the results about klu RNAi and miranda expression. If Klu is not expressed in ISCs, why is miranda altered (isn't it ISC specific?).

Minor:

The term EB is not defined- please write out "enteroblast" first. Also, it would be good to define it as the literature is becoming a bit confused. Are you using it as the "stem cell sister" or the "Notch active stem cell sister"? From your statement on p6 that "the Notch activity reporter Su(H)GBE-LacZ, which is exclusively activated in EBs, but not in ISCs", it sounds like the later definition. In which case in the intro when you say "differentiation of a subset of EBs into EEs" is not correct. It would be helpful to be more precise. Also, I feel that the description of "pre-determined ISCs" may be misleading as well. It sounds like there are two populations of ISCs one that makes EEs and one that makes ECs. The data argue that these cell types interconvert and are not really "predetermined".

-still a reference to "precise temporal regulation of N signaling" on p5. Again, to me no evidence of "temporal" is presented. They could simply move this statement after the "We propose..."

-For the new experiment on Su(H)GBE-F/O of Klu RNAi where Ecc15 was used, please indicate that

this is upon Ecc15 infection on the figure itself.

-add genotype of Su(H)F/O to methods

– p8 please state that this also did this in MARCM clones as it is not clear.

-P10- missing figure reference for statement “esg-F/O clones expressing UAS-Klu did not stain positive for either DI-lacZ or Su(H)GBE-lacZ”

Reviewer #2:

Remarks to the Author:

As before I am happy with the authors' response to my comments

Response to reviewers NCOMMS-18-13852A

Reviewer #1 (Remarks to the Author):

The authors have satisfactorily addressed a number of my points (1, 2, 4, 5, 9, 10, 12, 13, 14, and my minor points).

I still feel that there are two logical flaws in the interpretation of the data presented in this manuscript. The first has to do with the interpretation of the role of Klu overexpression in differentiation. If cell division is blocked, it follows that cell differentiation cannot occur.

- I do not feel that the authors addressed my point 3 regarding this. They still maintain throughout their entire section on p9 that this is affecting “differentiation”. I do not believe that they “adjusted the main text to include a statement about Klu expression impairing clonal proliferation”. They also did not add the PHH3 and DI staining and quantification that I suggested. Certainly, their unpublished data on a link with cell cycle and differentiation is interesting, but it does not impact the interpretations made in this manuscript.
- I still maintain that the authors cannot conclude on p9 that overexpression of Klu inhibits stem cell differentiation without at least saying in **the same sentence** that it blocks cell division. Of course, if there is no cell division, there cannot be any differentiation. The focus of the paragraph is on differentiation and division is not mentioned.
Presumably, the inhibition of cell division upon the overexpression of Klu is due to activity in the ISC (where it is not normally expressed). The physiological relevance of this to me is not clear. This being said, I do appreciate that their new data on Su(H)GBE-F/O klu RNAi do provide supporting evidence for an important role in Su(H)GBE+EBs to restrict EE fate. I would caution the authors to be careful in overinterpreting data.

We apologize for not being clear in the new version of our manuscript. We had included a statement regarding the inhibition of proliferation by Klu over-expression on page 10: *‘This suggests that Klu acts downstream of Notch in EB differentiation, but additionally acts as a potent inhibitor of cell proliferation’*, and on pages 11-12: *‘Reciprocally, ectopic Klu expression interferes with normal Delta-Notch signaling between ISC and EB and inhibits proliferation.’*

As requested, we have now also edited page 9, stating: *‘In contrast, clones expressing full-length Klu remained very small, containing only a few cells that did not exhibit any hallmarks of differentiation into either EEs or ECs (Figure 3B).’* and *‘Whereas clones grew normally and differentiation still occurred in clones expressing the activating Klu-VP16, clone size was smaller and differentiated cells were not observed in clones expressing the repressing Klu-ERD, confirming that transcriptional repression of genes regulated by Klu is sufficient to limit growth of ISC-derived clones (Figure 3C-D, quantification in E)’*. We have also changed the heading of this section on p9.

Furthermore, we highlight that the expression of Klu using *esg^{fs}* leads to a loss of mitoses in the ISC population. We had already included these data (in Fig. S3e), but we are now highlighting it by placing it in the main Fig. 3: quantification of pH3+ cells showed that UAS-klu (in a wild-type background) indeed negatively affects mitoses rates compared to wild-type control animals. We have made mention of this in the Main Text and now include statistical analysis between these groups (Control average= 4.2 mitoses/midgut, compare to UAS-klu: 0.48 mitoses/midgut, $P < 0,0001$, Figure 3E) to clarify this issue.

We agree that Klu is not expressed in ISCs under normal physiological conditions and that interpretation of the over-expression experiment (as for any gain-of-function experiment) therefore has to be approached with caution. We posit that the inhibition of ISC proliferation by the ectopic over-expression of Klu in ISCs provides evidence for a role of Klu in shutting down the mitotic cell cycle in EB's after the ISC-EB division/specification event. In combination with our data showing the repression of tumorigenesis in a N^{RNAi} or UAS-Ras^{v12} context, as well as our DamID data showing binding of Klu at several cell cycle genes, including CycE and CycB, we believe that such an interpretation is warranted. But we have edited the text to acknowledge the caveats.

With respect to the point that the inhibition of ISC proliferation precludes differentiation, we believe that we have not been clear enough when discussing differentiation. We wanted to convey that if Notch signaling induces Klu in EBs, and the loss of Klu leads to increased specification of EE, then the over-expression of Klu may conceivably prevent specification/differentiation of EEs and cause the forced differentiation of ISCs (even non-dividing ISCs) into enterocytes. Such a phenotype is observed, for example, when N-ICD is over-expressed in ISCs. Since we do not see such forced differentiation, our interpretation was that excessive Klu (or prolonged expression of Klu) inhibits differentiation (and thus Klu needs to be downregulated in EBs by its negative feedback loop). We fully agree that based on these experiments we cannot make a statement regarding a role for Klu in blocking EE differentiation. We have made substantial edits to this paragraph to clarify our interpretation.

The second problem that I see with the logic is regarding their interpretation of the role of Klu overexpression in Notch signaling: obviously if Delta expression is blocked, no downstream signaling can occur. I would say that this is one effect and not two, as they maintain "This suggests that Klu expression promotes an exit from the (DI+) stem cell state, **but also** interferes with transcriptional programs induced by Delta-Notch signaling in EBs." In addition, the loss of DI expression upon overexpression is likely occurring in the ISC where Klu is not normally expressed and again, how this relates to the normal physiological context where Klu is only expressed in EBs is not clear to me.

We agree with the reviewer that our interpretation of the data presented in Figure 3 and Supplementary Figure S3 at this point in the manuscript was premature. Interfering with Delta expression in ISCs (visualized by the loss of DI-lacZ in UAS-klu clones) would indeed lead to a loss of Notch activation (visualized by Su(H)GBE-lacZ) in the EB.

We further agree that Klu is not normally expressed in ISCs, but we believe that ectopic expression of Klu in ISCs does provide an additional line of evidence into how Klu functions to regulate the ISC > EB transition: namely, triggering exit from the mitotic cell cycle as well as repressing Delta expression.

Importantly, our model on Klu function in EBs does not solely depend on the ectopic expression data, but is further supported by the Klu loss-of-function, RNA-Seq and DamID data that indicate a role for Klu in modulating Notch target gene expression and in cell cycle control.

Yet, since we do concur with the reviewer that our interpretation of the data presented in this section of the manuscript was premature, we have removed the statement on the control of Notch-regulated transcriptional programs in this section. Instead, we state '*suggesting that ectopic Klu expression in ISCs interferes with normal DI-Notch signaling in ISC-EB pairs*' (p10 of the Revised Main Text).

-I am not sure to understand the results about *klu* RNAi and *miranda* expression. If *Klu* is not expressed in ISCs, why is *miranda* altered (isn't it ISC specific?).

The expression of the *mira*-Promoter-GFP construct as used in Figure 4 was reported to be ISC-specific in Bardin et al., *Development* 2010. However, closer examination shows that *mira*P-GFP expression largely overlaps with *Esg* expression, and is thus expressed both in ISCs and EBs (See Figure 6 F, Korzelius et al., 2014 EMBO Journal). To further illustrate this, we have added a close-up panel of the *mira*P-GFP expression pattern in ISC-EB nests in wild type intestines (Figure 4F) as Figure 4F'. ISCs can be distinguished from EBs in pairs of *Esg*+ cells based on nuclear size (Arrowheads and arrows, respectively), and *mira*P-GFP expression is seen in both. Critically, *mira*P-GFP expression is increased in EBs in *klu*^{RNAi} *Esg*+ cells, indicating that *Klu* suppresses *mira* expression in these cells. Furthermore, our *Klu* gain-of-function data indicate that over-expressing *Klu* ectopically in ISCs is sufficient to suppress *mira* expression even in ISCs. Hence, we propose that physiologically, the activation of *Klu* expression in EBs contributes to the downregulation of *mira* expression in EBs. We have clarified this in the text.

Minor:

The term EB is not defined- please write out “enteroblast” first. Also, it would be good to define it as the literature is becoming a bit confused. Are you using it as the “stem cell sister” or the “Notch active stem cell sister”? From your statement on p6 that “the Notch activity reporter Su(H)GBE-LacZ, which is exclusively activated in EBs, but not in ISCs”, it sounds like the later definition. In which case in the intro when you say “differentiation of a subset of EBs into EEs” is not correct. It would be helpful to be more precise. Also, I feel that the description of “pre-determined ISCs” may be misleading as well. It sounds like there are two populations of ISCs one that makes EEs and one that makes ECs. The data argue that these cell types interconvert and are not really “predetermined”.

We agree with the reviewer that recent results from the literature have shown that the EC-EE differentiation decision is more complex than previously thought and that these results contradict the old notion of the EB as a precursor for both the EE and EC. Our definition of the EB-state is indeed the latter: a Notch-active stem cell sister that is the precursor of the EC. We have further clarified this in the Introduction. We have also removed the statement on the “differentiation of a subset of EBs into EEs” in the N^{RNAi} phenotype, as there are no Su(H)GBE-positive EBs formed in N^{RNAi} clones. We have also removed the description of “pre-determined ISCs”, since we did not want to suggest the presence of 2 distinct ISC populations that are somehow intrinsically separate from each other.

Still a reference to “precise temporal regulation of N signaling” on p5. Again, to me no evidence of “temporal” is presented. They could simply move this statement after the “We propose...”

We have removed this statement from the revised Main text.

For the new experiment on Su(H)GBE-F/O of *Klu* RNAi where *Ecc15* was used, please indicate that this is upon *Ecc15* infection on the figure itself.

We have added this statement in the heading of Figure panels 2E-F.

Add genotype of Su(H)F/O to methods

We have added this information to the Methods section.

p8 please state that this also did this in MARCM clones as it is not clear.

We have corrected this error on page 8 in our revised Main Text

P10- missing figure reference for statement “esg-F/O clones expressing UAS-Klu did not stain positive for either DI-lacZ or Su(H)GBE-lacZ”

We have added the reference to Figure 3F-G (DI-lacZ) and Figure 3H-I (Su(H)GBE-lacZ) in our revised Main Text (p10).

--

Reviewer #2 (Remarks to the Author):

As before I am happy with the authors' response to my comments

We thank the reviewer for his/her time reviewing the manuscript.

Reviewers' Comments:

Reviewer #1:

Remarks to the Author:

The authors have now addressed my concerns.
thanks

Response to reviewers NCOMMS-18-13852C-Z

Reviewer #1 (Remarks to the Author):

The authors have now addressed my concerns.
thanks

We thank the reviewer for his/her time reviewing the manuscript.